# Constructing Non-isotropic Gaussian Diffusion Model Using Isotropic Gaussian Diffusion Model for Image Editing

**Xi Yu**[*], **Xiang Gu**[*], **Haozhi Liu, Jian Sun** (✉)
School of Mathematics and Statistics, Xi'an Jiaotong University, Xi'an, China
`{ericayu,xianggu,liuhzh}@stu.xjtu.edu.cn {jiansun}@xjtu.edu.cn`

## Abstract

Score-based diffusion models (SBDMs) have achieved state-of-the-art results in image generation. In this paper, we propose a Non-isotropic Gaussian Diffusion Model (NGDM) for image editing, which requires editing the source image while preserving the image regions irrelevant to the editing task. We construct NGDM by adding independent Gaussian noises with different variances to different image pixels. Instead of specifically training the NGDM, we rectify the NGDM into an isotropic Gaussian diffusion model with different pixels having different total forward diffusion time. We propose to reverse the diffusion by designing a sampling method that starts at different time for different pixels for denoising to generate images using the pre-trained isotropic Gaussian diffusion model. Experimental results show that NGDM achieves state-of-the-art performance for image editing tasks, considering the trade-off between the fidelity to the source image and alignment with the desired editing target.

## 1 Introduction

Score-based diffusion models (SBDMs) [1–6] demonstrate state-of-the-art performance on image synthesis quality and sample diversity. SBDMs are widely applied to applications such as text-to-image synthesis [7–9], image editing [10–15], deblurring [16, 17], etc. SBDMs consist of a forward diffusion stage that adds random noise to data and a reverse stage that generates desired data from noise. The introduced noise in the forward process is commonly isotropic Gaussian noise [1, 6], i.e., independently and identically distributed noise in a normal distribution.

Non-isotropic diffusion model, by adding non-isotropic noises in the forward diffusion process has been investigated in [18–22]. The blurring diffusion model in [18] adds blur and noise to samples which is a Gaussian diffusion process with non-isotropic noise in the frequency space. [19] employs auxiliary velocity variables to augment the data variables as Hamiltonian dynamics, and performs the diffusion process in this expanded joint space by adding different noises to the auxiliary and the data variables. [20] formulates the diffusion model using non-isotropic noise with a positive semi-definite covariance matrix, and carries out a comparative analysis of the non-isotropic and isotropic diffusion models. These works have shown better performance for data generation.

This paper focuses on image editing tasks that commonly require editing specific object/thing of an image while preserving the remaining parts of the image. For image editing, [11] produces a mask that allows the preservation of context while editing the remaining part. [23–25] use the learned text embedding for the object that needs to be preserved in the image to ensure the object is unchanged during editing. It is empirically known that the diffusion model can generate more diverse novel

---

* Equal contribution.

37th Conference on Neural Information Processing Systems (NeurIPS 2023).

content if adding noise with larger variance to the image while preserving the image content if adding smaller variance noise [13]. Motivated by this, we employ a non-isotropic diffusion model to add noises with different variances to different image pixels, considering the degree to which the corresponding pixels should be edited/preserved.

Along with this idea, we present a Non-isotropic Gaussian Diffusion Model (NGDM), utilizing an off-the-shelf isotropic diffusion model (e.g., DDPM [1]) for achieving the data sampling in image editing. Specifically, given a source image, the proposed NGDM is with a diffusion process that adds independent Gaussian noises with different variances to different pixels, therefore the added noise is independent and non-isotropic over image pixels. We then employ an off-the-shelf isotropic diffusion model to execute the reverse denoising process for image editing. To achieve this goal, we rectify the NGDM into the isotropic Gaussian diffusion model where each pixel is added with the same amount of noise at each step but different pixels have varying total number of noise accumulation time steps. We subsequently devise a specific sampling method for NGDM that can generate images by using the pre-trained isotropic Gaussian diffusion model.

We demonstrate the effectiveness of NGDM in image editing tasks on five datasets including real and synthetic datasets. In the experiments for cats to dogs editing task on AFHQ dataset, our method achieves the best performance in the metric of FID and SSIM compared with the SoTA SBDMs-based methods. For text-guided image editing, our method achieves better trade-off between CLIPScore and LPIPS value. Furthermore, NGDM allows for flexible trade-off with varying hyper-parameters.

## 2 Background: Score-based Diffusion Models

SBDMs [1, 2, 4–6] are a family of generative models that learn the data distribution based on the Gaussian process. Two representative models are Denoising Diffusion Probabilistic Model (DDPM) [1] and Score Matching with Langevin Dynamics (SMLD) [5]. We discuss the details based on DDPM for the remainder of the main text for brevity.

Given the input data $\mathbf{x}(0) \in \mathbb{R}^D$, which represents a sample from the data distribution $p_{data}$, a forward process produces the noisy $\mathbf{x}(t)$ indexed by a time variable $t \in [0, 1]$ via

$$\mathbf{x}(t) = \sqrt{\bar{\alpha}(t)}\mathbf{x}(0) + \sqrt{1 - \bar{\alpha}(t)}\mathbf{z}(t), \tag{1}$$

where $\mathbf{z}(t) \in \mathcal{N}(\mathbf{0}, \mathbf{I})$ for any $t$ and $\bar{\alpha}(t) = e^{-\int_0^t \beta(s)\mathrm{d}s}$ controlling the noise schedule. $\beta(s) = \bar{\beta}_{\min} + s(\bar{\beta}_{\max} - \bar{\beta}_{\min})$ with $\bar{\beta}_{\min} = 0.1$ and $\bar{\beta}_{\max} = 20$ [1, 6]. This type of diffusion model is dubbed *Isotropic Gaussian Diffusion Model (IGDM)*, since the added Gaussian noise $\mathbf{z}(t)$ is from the independently and identically distributed normal distribution.

DDPM is in the framework of Stochastic Differential Equations (SDEs) [5] with variance preservation

$$\mathrm{d}\mathbf{x}(t) = -\frac{1}{2}\beta(t)\mathbf{x}(t)\mathrm{d}t + \sqrt{\beta(t)}\mathrm{d}\mathbf{w} \quad \text{with initial value } \mathbf{x}(0), \tag{2}$$

where $\mathbf{w}$ is the standard Wiener process. The reverse process denoises the noisy sample $\mathbf{x}(T)$ starting from $T$ using a reverse SDE

$$\mathrm{d}\mathbf{x}(t) = \left[-\frac{1}{2}\beta(t)\mathbf{x}(t) - \beta(t)\nabla_{\mathbf{x}}\log p_t(\mathbf{x}(t))\right]\mathrm{d}t + \sqrt{\beta(t)}\mathrm{d}\bar{\mathbf{w}} \quad \text{with initial value } \mathbf{x}(T), \tag{3}$$

where $\bar{\mathbf{w}}$ is a standard Wiener process when time flows backward from $T$ to $0$. The score function $\nabla_{\mathbf{x}}\log p_t(\mathbf{x})$ is approximated by training a time-dependent model $\mathbf{s}_{\boldsymbol{\theta}}(\mathbf{x}(t), t)$ via score matching [6, 26]. For inference, the time of the differential equation is discretized as $t \in \{0, \Delta t, 2\Delta t, \cdots, T\}$ with $\Delta t$ representing the sampling time interval. We can choose to utilize the reverse process of DDPM or the reverse process of DDIM for sampling. With $\beta_t = \beta(t)\Delta t$, the iteration rule of DDPM [1] is

$$\mathbf{x}(t) = \frac{1}{\sqrt{1 - \beta_{t+\Delta t}}}\left(\mathbf{x}(t + \Delta t) + \beta_{t+\Delta t}\mathbf{s}_{\boldsymbol{\theta}}\left(\mathbf{x}(t + \Delta t), t + \Delta t\right) + \sqrt{\beta_{t+\Delta t}}\mathbf{z}(t + \Delta t), \tag{4}$$

where $\mathbf{z}(t + \Delta t) \in \mathcal{N}(\mathbf{0}, \mathbf{I})$. With $\bar{\alpha}_t = \prod_{s=0}^t (1 - \beta_s)$, the iteration rule of DDIM [27] is

$$\mathbf{x}(t) = \sqrt{\bar{\alpha}_t}\left(\frac{\mathbf{x}(t + \Delta t) + (1 - \bar{\alpha}_{t+\Delta t}) \cdot \mathbf{s}_{\boldsymbol{\theta}}(\mathbf{x}(t + \Delta t), t + \Delta t)}{\sqrt{\bar{\alpha}_{t+\Delta t}}}\right)$$
$$- \sqrt{(1 - \bar{\alpha}_t - \sigma^2(t + \Delta t))(1 - \bar{\alpha}_{t+\Delta t})} \cdot \mathbf{s}_{\boldsymbol{\theta}}\left(\mathbf{x}(t + \Delta t), t + \Delta t\right) + \sigma(t + \Delta t)\mathbf{z}(t + \Delta t). \tag{5}$$

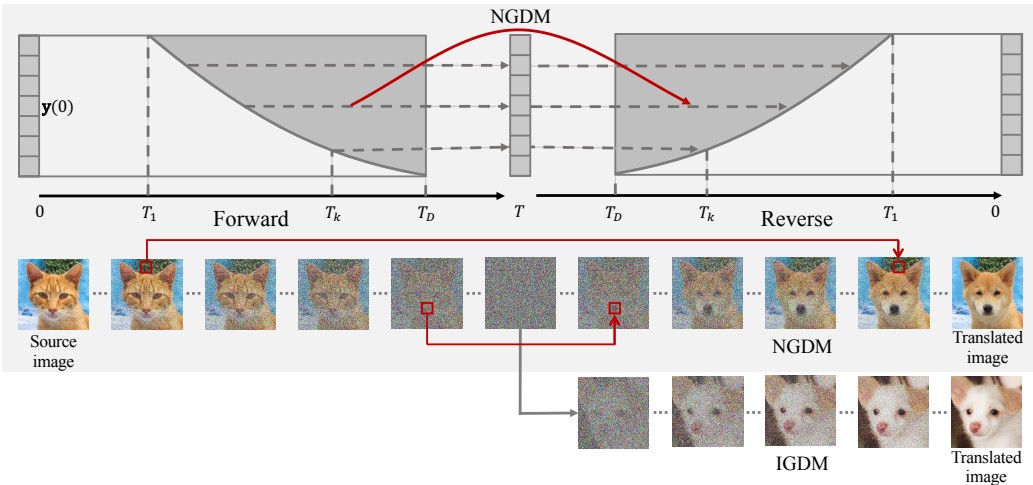

Figure 1: The overview of our NGDM. We rectify the non-isotropic diffusion model into isotropic model with different total time steps (e.g., $T_1, \cdots, T_D$) for different pixels. Based on this rectification, the input data $\mathbf{y}(0)$ is firstly added isotropic noises until $T$ time steps. Then in the reverse denoising process, to apply the pre-trained isotropic Gaussian diffusion model, we construct the noisy image to be denoised in each time step following Eq. (12). The red arrow in the figure indicates the pixel replacement operation in Eq. (12) when $T_k \leq t \leq T$ for the $k$-th pixel at a denoising time step $t$.

## 3   Method

In this section, we present a framework for utilizing the pre-trained Isotropic Gaussian Diffusion Model (IGDM) to achieve the sampling process of Non-isotropic Gaussian Diffusion Model (NGDM). In the following, first, we define the NGDM with added independent non-isotropic Gaussian noise. Then, we implement NGDM by IGDM through rectifying the spatially different time of noising and denoising procedures in NGDM and present our proposed data sampling algorithm for the proposed NGDM using the pre-trained IGDM.

### 3.1   Non-isotropic Gaussian Diffusion Model

In this work, we focus on the Non-isotropic Gaussian Diffusion Model (NGDM) by adding the non-isotropic Gaussian noise in the input data $\mathbf{y}(0) \in \mathbb{R}^D$ and $\mathbf{y}(0) \sim p_{data}$, and the noises associated with different pixels are independent. The forward SDE of NGDM is

$$d\mathbf{y} = -\frac{1}{2}\beta(t)\mathbf{\Lambda}(\mathcal{I})\mathbf{y}dt + \sqrt{\beta(t)\mathbf{\Lambda}(\mathcal{I})}d\mathbf{w} \quad \text{with initial value } \mathbf{y}(0), \tag{6}$$

where $\mathcal{I} \in \mathbb{R}^D$ is the source data, $\mathbf{\Lambda}(\mathcal{I}) : \mathbb{R}^D \to \mathbb{R}^{D \times D}$ is the weighting coefficient matrix, defined as diagonal matrix $\mathbf{\Lambda}(\mathcal{I}) = \mathrm{diag}\left(\lambda_1, \cdots, \lambda_D\right)$ with $0 \leq \lambda_k \leq 1$ scaling the Gaussian noise level added to the $k$-th pixel. Note that the transition kernel $p_{0t}(\mathbf{y}(t)|\mathbf{y}(0)) = \mathcal{N}\left(\mathbf{y}(t) \mid \mathbf{y}(0)e^{-\frac{1}{2}\int_0^t \beta(s)\mathbf{\Lambda}(\mathcal{I})ds}, \mathbf{I} - e^{-\int_0^t \beta(s)\mathbf{\Lambda}(\mathcal{I})ds}\right)$ is an independent Gaussian distribution.

### 3.2   Rectify the Non-isotropic Gaussian Diffusion Model

With the added independent noise, we next discuss the forward SDE for NGDM in scalar form for each pixel $k$. Given the initial $\mathbf{y}^k(0)$ denoting the value of pixel $k$ in $\mathbf{y}(0)$, the forward SDE of the $k$-th pixel can be presented by

$$d\mathbf{y}^k = -\frac{1}{2}\beta(t)\lambda_k\mathbf{y}^k dt + \sqrt{\beta(t)\lambda_k}d\mathrm{w} \quad \text{with initial value } \mathbf{y}^k(0), \tag{7}$$

where w is a one-dimensional Wiener process.

We present Theorem 1 to illustrate the connection between the NGDM defined in Section 3.1 and the IGDM defined in Section 2 at the pixel level. Beforehand, we introduce the following lemma.

**Lemma 1.** *Let $\beta(s) = \bar{\beta}_{\min} + s(\bar{\beta}_{\max} - \bar{\beta}_{\min})$ with $\bar{\beta}_{\max} > \bar{\beta}_{\min} > 0$. Then, for each $\lambda_k \in [0, 1]$ and $t \in [0, 1]$, there exists a unique time $\tau \in [0, 1]$ (denoted by $\tau = \xi_k(t)$) such that $\int_0^t \beta(s)\lambda_k \mathrm{d}s = \int_0^\tau \beta(s)\mathrm{d}s$ and $\beta(t)\lambda_k \mathrm{d}t = \beta(\tau)\mathrm{d}\tau$, with the following form*

$$\xi_k(t) = \frac{-\bar{\beta}_{\min} + \sqrt{\bar{\beta}_{\min}^2 + 2(\bar{\beta}_{\max} - \bar{\beta}_{\min})\bar{\beta}_{\min} t \lambda_k + (\bar{\beta}_{\max} - \bar{\beta}_{\min})^2 t^2 \lambda_k}}{\bar{\beta}_{\max} - \bar{\beta}_{\min}}. \tag{8}$$

The proof is in Appendix A. Based on the above Lemma, we can rectify the NGDM, which adds noise at each pixel with varying variance over the same time span, into an IGDM that adds noise at each pixel with the same noise variance but with different total diffusion time for different pixels. We introduce the following theorem to derive the differential equation as an IGDM.

**Theorem 1.** *For a pixel indexed by $k$, $\lambda_k \in [0, 1]$, and let $\tau = \xi_k(t)$ with $\xi_k(t)$ represented in Eq. (8). With the same initial value $\mathbf{y}^k(0)$, we have that the transition kernel at time $t$ induced by Eq. (7) is equal to the transition kernel at time $\tau$ induced by the following differential equation*

$$\mathrm{d}\mathbf{y}^k = -\frac{1}{2}\beta(\tau)\mathbf{y}^k \mathrm{d}\tau + \sqrt{\beta(\tau)}\mathrm{d}\mathrm{w} \quad \text{with initial value } \mathbf{y}^k(0). \tag{9}$$

*The total time of noising for Eq. (9) is $T_k$ with $T_k = \xi_k(T)$.*

We provide the proof in Appendix A. Inspired by this, we rectify the reverse process in NGDM with different speeds of denoising across pixels to be the reverse process with consistent speed but different total time of denoising. We suggest rectifying the differential equation for the reverse process of pixel $k$ within the NGDM framework into the following form

$$\mathrm{d}\mathbf{y}^k = \left[-\frac{1}{2}\beta(\tau)\mathbf{y}^k - \beta(\tau)(\nabla_{\mathbf{y}} \log p_\tau(\mathbf{y}))^k\right]\mathrm{d}\tau + \sqrt{\beta(\tau)}\mathrm{d}\bar{\mathrm{w}} \quad \text{with initial value } \mathbf{y}^k(T_k), \tag{10}$$

where $\bar{\mathrm{w}}$ is a one-dimensional Wiener process when time flows backward from $T_k$ to $0$. Theorem 1 establishes the conclusion that the NGDM in Eq. (7) can be rectified to the IGDM in Eq. (9) but with different total diffusion time $T_k$ for different pixel indexed by $k$, determined based on Eq. (8). This inspires us to utilize the pre-trained IGDM to achieve the data sampling of NGDM for image editing. Subsequently, we present a method that adjusts the total time of noising and denoising for each pixel $k$ to $T_k$, enabling us to use the pre-trained IGDM for data sampling. The corresponding sampling method is presented in Algorithm 1.

For image editing tasks, we use the source image $\mathcal{I}$ as $\mathbf{y}(0)$ and generate noisy data $\mathbf{y}(T)$ through the forward process. We generate the edited image $\hat{\mathbf{y}}(0)$ by denoising from $\mathbf{y}(T)$. Utilizing the forward noising process of IGDM, we add independent noise to each pixel $k$ to obtain the noisy observation $\mathbf{x}^k(t)$ of discrete time $t \in \{0, \Delta t, \cdots, T\}$ with $\Delta t$ representing the sampling time interval

$$\mathbf{x}^k(t) = \sqrt{\bar{\alpha}_t}\mathcal{I}^k + \sqrt{1 - \bar{\alpha}_t}\mathbf{z}^k(t), \tag{11}$$

where $\mathbf{z}(t) \in \mathcal{N}(\mathbf{0}, \mathbf{I})$. Next, with $\mathbf{H}(\mathbf{y}(t + \Delta t), t + \Delta t)$ denotes the sampling procedure of DDPM and DDIM given in Eq. (4) and Eq. (5) of Section 2, the data sampling iteration utilizing the IGDM model with initial value $\mathbf{y}^k(T) = \mathbf{x}^k(T)$ is defined as

$$\mathbf{y}^k(t) = \begin{cases} \mathbf{x}^k(t) & T_k \le t < T \\ \mathbf{H}^k(\mathbf{y}(t + \Delta t), t + \Delta t) & \text{otherwise.} \end{cases} \tag{12}$$

This implies that we use the noisy observation $\mathbf{x}^k(t)$ to represent $\mathbf{y}^k(t)$ at each step before $T_k$ with $t \ge T_k$, rather than the actual denoised result starting from time step $T$. Until time $T_k$ we begin the denoising from $\mathbf{x}^k(T_k)$ for $k$-th pixel. In such a way, different pixels have different starting time steps ($T_k$ for $k$-th pixel) for image denoising in the data sampling process.

### 3.3 Sampling Method in NGDM

Based on the above method, we further specify our sampling algorithm by utilizing a pre-trained IGDM. We do not require the training of NGDM, but instead harness the power of a pre-trained IGDM for data sampling. We generate the edited image with the source image $\mathcal{I}$ as a condition. We first add noise to the source image $\mathcal{I}$ to $T$ time steps as the starting point of denoising, and then use the method in Section 3.2 to rectify NGDM into IGDM to denoise the image using the pre-trained IGDM. We show the sampling algorithm of NGDM in the Algorithm 1.

---

**Algorithm 1** Sampling Method of NGDM

---

**Inputs:** The source image $\mathcal{I}$, the weighting matrix $\mathbf{\Lambda}(\mathcal{I})$, the score function $s_{\boldsymbol{\theta}}$, the time schedule $\{\beta(t)\}_{t=0}^{T}$, the maximal time steps $T$
1: Compute $\mathbf{T} = [\xi_1(T), \cdots, \xi_D(T)]$ according to Eq. (8)
2: $\mathbf{z} \sim \mathcal{N}(\mathbf{0}, \mathbf{I})$
3: $\mathbf{y} = \sqrt{\bar{\alpha}_T}\mathcal{I} + \sqrt{1 - \bar{\alpha}_T}\mathbf{z}$      # The starting point of denoising
4: **for** $t = T$ to 0 **do**
5:     $\mathbf{x} = \sqrt{\bar{\alpha}_t}\mathcal{I} + \sqrt{1 - \bar{\alpha}_t}\mathbf{z}$      # Sample the noisy source image at time step $t$
6:     $\mathbf{M} = \mathbb{I}(t \geq \mathbf{T}\})$
7:     $\mathbf{y} \leftarrow \mathbf{M} \odot \mathbf{x} + (\mathbf{1} - \mathbf{M}) \odot \mathbf{y}$     # $\mathbf{1}$ is the $D$-dimensional all-ones vector
8:     $\mathbf{y} \leftarrow \mathbf{H}(\mathbf{y}, t)$
9: **end for**
**Output:** Generated image $\mathbf{y}$ conditioned on the source image $\mathcal{I}$

---

## 3.4   Design of Input-dependent Weighting Matrix

We construct the weighting matrix $\mathbf{\Lambda}(\mathcal{I})$ based on the method for designing mask using a text-conditioned diffusion model [9] in DiffEdit [11]. Specifically, given the source image $\mathcal{I}$, the text description $R$ of the source image, and the target description $Q$ that describes the desired target image after editing. Following DiffEdit [11], we add noise to the source image up to $0.5T$ step and use the texts $R$ and $Q$ as the conditions respectively for denoising in the current time step to estimate the score values by using the score network $\mathbf{s}_{\boldsymbol{\theta}}$. We derive an attention map $\mathcal{A}(\mathcal{I})$ based on the absolute difference of the estimated scores. We use the above method to compute 10 estimated absolute noise differences by running 10 times with different random seeds, averaging and performing Gaussian smoothing on the averaged map. Finally, we normalize the values of the smoothed map to $[0, 1]$ as the final attention map $\mathcal{A}(\mathcal{I})$. The pixel with larger value in the attention map should be added with the noise with larger variance, to sufficiently edit the content of the pixel. The pixel with smaller value in the attention map should be added with noise having smaller variance to preserve the content of pixel. We transform the attention map $\mathcal{A}(\mathcal{I})$ into the weighting matrix $\mathbf{\Lambda}(\mathcal{I})$ through a Sigmoid function by $\mathbf{\Lambda}(\mathcal{I}) = \frac{1}{1+\exp(-(a\mathcal{A}(\mathcal{I})-b))}$, where $a$ and $b$ are the hyper-parameters for the transformation. We discuss the effect of hyper-parameters $a$ and $b$ on the generation of images in Section 4.

# 4   Experiment

In this section, we apply the proposed NGDM to image editing tasks, presenting both qualitative and quantitative results.

## 4.1   Experimental Setup

**Evaluation tasks.**   We perform experiments on five datasets. First, we experiment on AFHQ dataset [28] to edit cats into dogs. The source domain contains 500 images of cats. These images are resized to $256 \times 256$ resolution and subsequently edited into dogs. Second, we experiment on ImageNet dataset [29] to edit images from one class into another class based on text prompts, following the protocol of [30]. Third, we experiment on synthetic Imagen [31] dataset following DiffEdit [11]. We collect images generated by Imagen [31], along with their corresponding text prompts, and edit these images by altering portions of the text. Fourth, we experiment on COCO-S dataset and construct target prompts for editing from annotations provided by BISON [32]. We collect 1000 images and target prompts from the COCO [33] dataset to build the COCO-S dataset. We additionally consider DreamBooth dataset provided by [24], which contains 30 objects. Each object has 25 prompts and 4-6 images for editing. We edit each image using the provided 25 prompts, resulting in a total of 3950 edited images. More details are available in Appendix B.1.

**Implementation details.**   We conduct experiments utilizing two types of pre-trained diffusion models. For the cats to dogs editing task on AFHQ dataset, we utilize the public pre-trained score-based diffusion model with the official code provided in ILVR [10]. This model operates directly in the image space. We set the denoising step $N$ to 1000. We implement the remaining task based on the text-to-image latent diffusion model, i.e., Stable Diffusion [9]. This model was pre-trained on 512

× 512 images of LAION dataset [34] with a latent dimension of 64 × 64. We use 50 steps in DDIM sampling with a fixed noise schedule. For hyper-parameters $a$ and $b$, we set them to 10.0 and 5.0 respectively to obtain all qualitative comparison results presented in this paper. We conduct additional analysis to investigate the effect of different values of hyper-parameters in the experimental results. Unless otherwise stated, we use the default parameters of the diffusion model during inference.

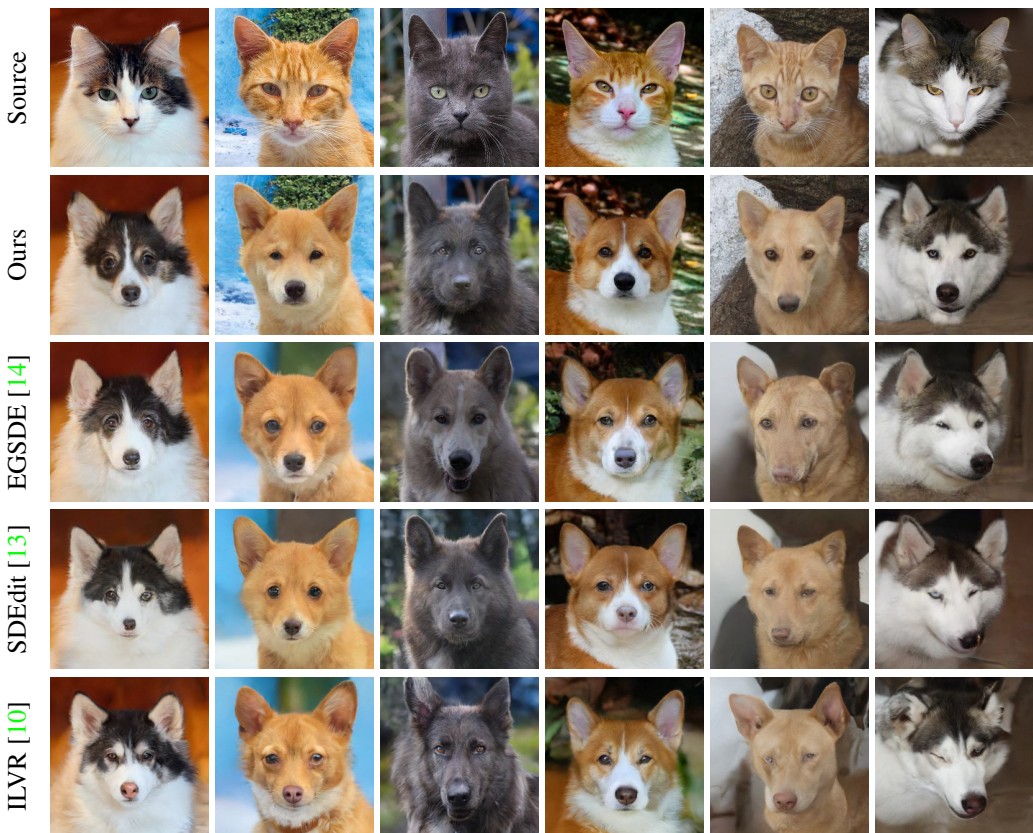

Figure 2: Qualitative comparison on AFHQ dataset.

## 4.2 Results and Analysis

**Results on AFHQ dataset.** We report the widely-used Frechet Inception Distance (FID) [35] for quantifying realism and SSIM [36] for quantifying faithfulness. The quantitative comparisons and qualitative results are presented in Table 1 and Figure 2, respectively. The images generated by our method better preserves the contextural structure (*e.g.*, pose) of cats while yielding realistic dog images. For example, Figure 2 shows that for images in columns 2-5 with complex backgrounds, our method can accurately keep the backgrounds unchanged while editing cats into dogs. Other methods either blur the backgrounds or fail to maintain the source image backgrounds correctly. This observation is further supported by the quantitative results in Table 1, where our method achieves the best results of FID and SSIM among the compared SBDM-based methods.

Table 1: Quantitative comparison on AFHQ dataset. All results are reported by repeating experiments 5 times.

| Method | FID ↓ | SSIM ↑ |
|---|---|---|
| StarGAN v2 [28] | 54.88 ± 1.01 | 0.27 ± 0.003 |
| CUT [37] | 76.21 | **0.601** |
| ILVR [10] | 74.37 ± 1.55 | 0.363 ± 0.001 |
| SDEdit [13] | 74.17 ± 1.01 | 0.423 ± 0.001 |
| EGSDE [14] | 65.82 ± 0.77 | 0.415 ± 0.001 |
| SDDM [38] | 62.29 ± 0.63 | 0.422 ± 0.001 |
| **Ours** | **61.39 ± 0.27** | 0.478 ± 0.001 |

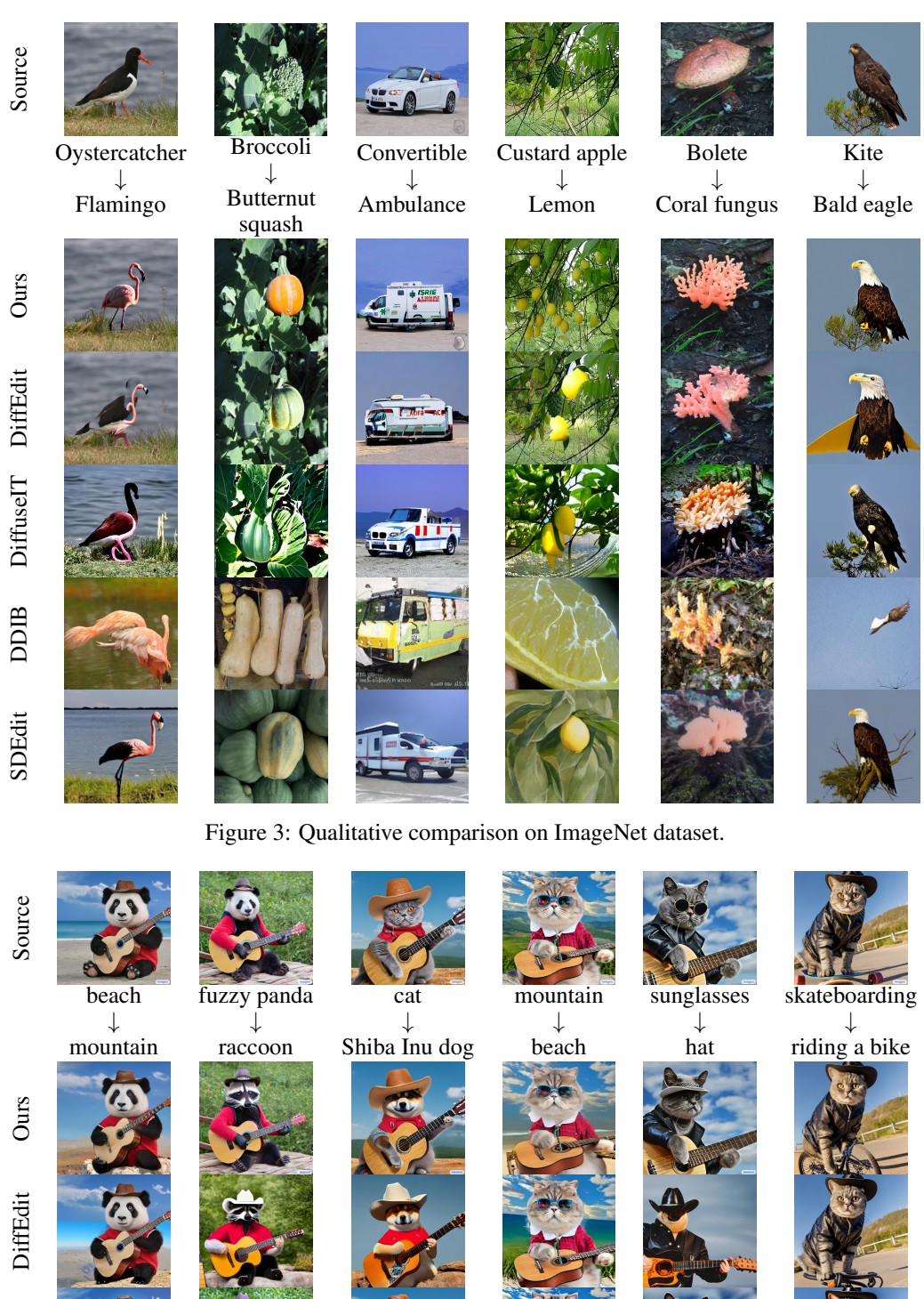

Figure 3: Qualitative comparison on ImageNet dataset.

Figure 4: Qualitative comparison on Imagen dataset.

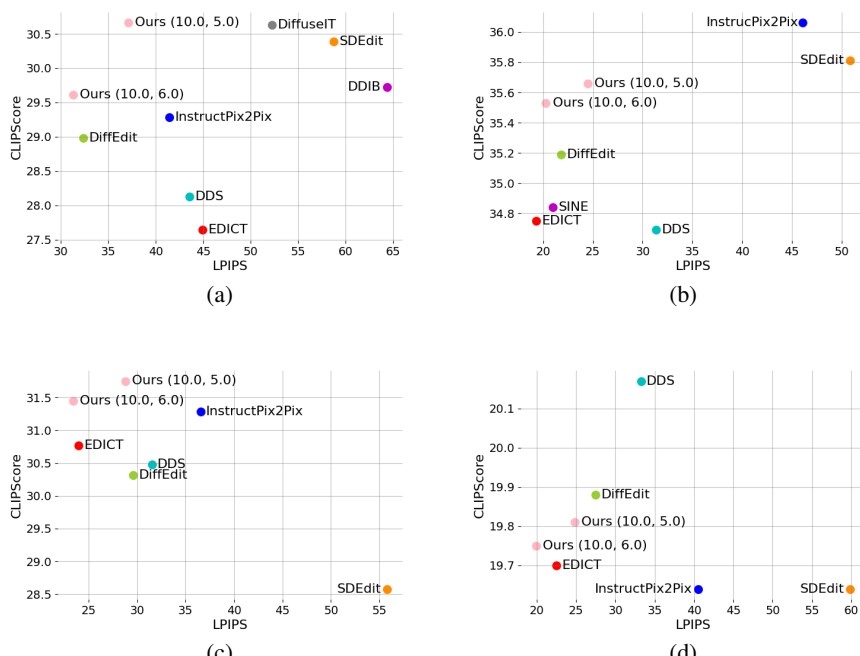

Figure 5: Quantitative comparison on ImageNet (a), Imagen (b), COCO-S (c), and DreamBooth (d) datasets. We report LPIPS distance [42] measuring image fidelity and CLIPScore [43] for text alignment. A higher CLIPScore denotes better alignment with the text, while a lower LPIPS value suggests higher fidelity to the input image. We report our results respectively using the default parameters $a = 10.0, b = 5.0$, and the parameters $a = 10.0, b = 6.0$.

**Results on ImageNet dataset.** Figure 3 shows that our method performs well on images even with complex backgrounds. For instance, when editing "custard apple" into "lemon", our generated image successfully preserves the intricate details of the tree branches. DiffEdit [11] generates unnatural images with artifacts. DiffuseIT [39] generates unnatural images, while DDIB [40] can hardly maintain the content of the source image. Figure 5(a) shows that our method outperforms other methods by achieving a better trade-off between CLIPScore and LPIPS value.

**Results on Imagen dataset.** We present our qualitative and quantitative results in Figures 4 and 5(b), respectively. Figure 4 shows visual results including background replacement and object properties modification. We can see that our method can generate images with better visual quality compared with the other methods. For instance, our results can successfully preserve the foreground while replacing the "beach" in the background with "mountain", or vice versa.

**Results on COCO-S dataset.** From Figure 5(c), when $a = 10.0$ and $b = 6.0$, our method achieves the highest CLIPScore 31.45, and the smallest LPIPS value 23.43 compared with all other methods. InstructPix2Pix [41] is able to obtain CLIPScore comparable to that of ours, but the LPIPS value of InstructPix2Pix [41] is worse. We provide qualitative comparison in Appendix B.2.

**Results on DreamBooth dataset.** Figure 5(d) shows the quantitative results on DreamBooth dataset. As can be observed, our method outperforms compared methods by achieving a better trade-off between CLIPScore and LPIPS value. DDS [44] achieves the best CLIPScore of 20.17 with a worse LPIPS value of 33.26. Our method obtains CLIPScore comparable to that of DiffEdit with better LPIPS value. We provide qualitative comparison in Appendix B.2.

Table 2: User study results on AFHQ dataset.

| ILVR [10] | SDEdit [13] | EGSDE [14] | Ours |
|-----------|-------------|------------|----------|
| 11.5%     | 10.5%       | 12.5%      | **65.5%** |

Table 3: User study results on ImageNet, Imagen, COCO-S, and DreamBooth datasets.

| SDEdit [13] | DiffEdit [11] | DDS [44] | EDICT [45] | InstructPix2Pix [41] | Ours |
|---|---|---|---|---|---|
| 4.5% | 10.0% | 3.0% | 4.5% | 6.0% | **72.0%** |

**User study.** We conduct two user studies on AFHQ dataset and the remaining datasets including ImageNet, Imagen, COCO-S, and DreamBooth. For each user study, 40 participants are provided with 30 randomly selected source images and the corresponding generated results of different methods. The generated images of ours and the other methods are displayed randomly in order. Participants are suggested to select the image that better applies the requested edit while preserving most of the original image details. The percentages of votes for

Table 4: The performance on AFHQ dataset with varying values of $a$ or $b$ while respectively fixing $b = 5.0$ or $a = 10.0$.

| $a$ ($b = 5.0$) | 6.0 | 8.0 | 12.0 | 14.0 |
|---|---|---|---|---|
| FID $\downarrow$ | 87.01 | 72.43 | 54.10 | 46.75 |
| SSIM $\uparrow$ | 0.556 | 0.513 | 0.449 | 0.425 |
| $b$ ($a = 10.0$) | 3.0 | 4.0 | 6.0 | 7.0 |
| FID $\downarrow$ | 42.34 | 50.35 | 74.58 | 88.49 |
| SSIM $\uparrow$ | 0.373 | 0.423 | 0.539 | 0.601 |

our method and the other method on different datasets are shown in Tables 2 and 3 respectively, demonstrating that the participants exhibit a strong preference towards our method.

**Effect of hyper-parameters $a$ and $b$.** As mentioned in Section 3.4, we transform the attention map $\mathcal{A}(\mathcal{I})$ into weighting matrix $\mathbf{\Gamma}(\mathcal{I})$ with hyper-parameters $a$ and $b$. We control the initial time step of denoising each pixel by adjusting the hyperparameters $a$ and $b$. In Table 4, we analyze the impact of hyper-parameters $a$ and $b$ on AFHQ dataset. The upper part of the table reports the results with varying $a$ and fixed $b = 5.0$, and the lower part of the table shows results with varying $b$ and fixed $a = 10.0$. With $b$ held constant, a larger $a$ results in higher level of noise to the image, leading to more accurate editing and obtaining smaller FID. A smaller $a$ results in content preserving the original image, obtaining larger SSIM value. The behavior of $b$ is opposite to $a$.

**Comparison with hard weighting matrix.** We compare with the strategy used to produce $\mathbf{\Lambda}(\mathcal{I})$ by $\mathbf{\Lambda}(\mathcal{I}) = \mathbb{I}(\mathcal{A}(\mathcal{I}) \geq \eta)$, where $\eta$ is a threshold chosen in $\{0.1, 0.3, 0.5, 0.7, 0.9\}$. The generated weighting matrix by this strategy is a "hard" weighting matrix that only takes 0 or 1 entry. Our Algorithm 1 gradually increases the denoising region with the increase of the denoising steps in the diffusion. Each pixel begins to be denoised with the denoising time step according to its relevance to the editing task. This helps to avoid artifacts caused by a hard mask, as shown in Figure 7.

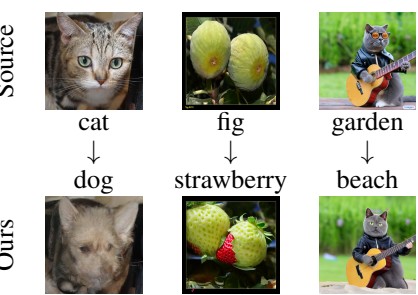

Figure 6: Failure examples. We show cases in which our method fails to generate high-quality edited results.

**Failure examples** We show several failure examples in Figure 6 that were unsuccessfully edited. This could be because the computed weighting matrix is not accurate for determining the scale of noise to be added to each pixel in the source image.

## 5 Related Work

Image editing aims to modify a real image to generate the desired image, resulting in tasks including image translation [46], style transfer [47], inpainting [48], object modification [24], etc. We focus on image editing tasks that commonly require editing specific object/thing of an image while preserving the remaining parts of the image.

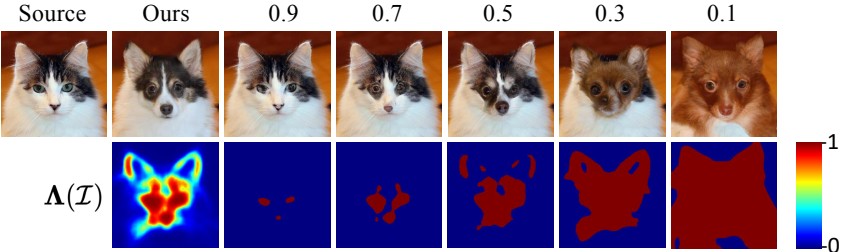

Figure 7: Edited images and heatmaps with soft and hard weighting matrix. The images in the second column represent the results generated by our method and the heatmap below the image depicts the weighting matrix $\mathbf{\Lambda}(\mathcal{I})$ defined in Section 3.1 in the paper. The images in columns 3-7 represent the results generated using the hard weighting matrix with threshold value in $\{0.1, 0.3, 0.5, 0.7, 0.9\}$. The heatmaps below the images represent the binary hard weighting matrix.

Diffusion models showcase remarkable results for image editing. SDEdit [13] employs the source noisy data as the starting point in the denoising process, and it explores the trade-off between realism and faithfulness by controlling the initialization of the denoising time. EGSDE [14] utilizes the energy functions trained on both source and target domains to guide the inference process. EGSDE uses the noisy data at $0.5T$ time step as the starting point of denoising, where $T$ denotes the total number of denoising time steps in DDPM. Compared with them, our derived sampling method incorporates an adaptive selection of the initial denoising time for each pixel during the denoising process. Methods of [12, 23, 24, 49] utilize text-conditioned diffusion models to fine-tune the text embedding of the object that needs to be preserved during editing using a few or a single image. DDS [44] utilizes delta scoring to provide effective gradients for editing. DiffEdit [11] uses the DDIM inversion method to obtain noisy data and automatically generates a binary mask to guide the denoising process. RePaint [50] tackles the inpainting task by taking the unmasked image region from the input image and the masked region from the DDPM generated image, using the hard 0-1 mask. Differently, we perform image editing by adding independent noise with different variances to different pixels, depending on a weighting coefficient matrix that contains soft weights. Pixels with less added noise will better preserve the content of the source image.

## 6 Conclusion, Limitations and Societal Impact

In this paper, we propose a Non-isotropic Gaussian Diffusion Model (NGDM) for image editing tasks. The NGDM adds independent Gaussian noises with varying variances to different image pixels. To avoid training score model for NGDM, we rectify NGDM into an isotropic Gaussian diffusion model and design a data sampling method for NGDM by using a pre-trained isotropic Gaussian diffusion model to generate images. We demonstrate that NGDM better trade-off the balance of realism and faithfulness than the state-of-the-art methods for image editing tasks.

A limitation of our method could be that incorrect weighting matrix may lead to the failure of the method. Moreover, our method relies on a pre-trained diffusion model. Artifacts are produced when the edit involves generation failure cases of the underlying model. In future work, we will design a better way to calculate the weighting matrix more precisely and efficiently. And we will explore the application of our method in downstream tasks, such as domain adaptation.

In our experiments, all the considered datasets are open-sourced and publicly available. Our work aims to manipulate images with minimum effort. However, this method might be misused by faking images. We will take care to exploit the method to avoid the potential negative social impact and we will help research in identifying and preventing malicious editing.

## 7 Acknowledgement

This work was supported by National Key R&D Program 2021YFA1003002, and NSFC (12125104, U20B2075,11971373).

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
