# Constructing Non-isotropic Gaussian Diffusion Model Using Isotropic Gaussian Diffusion Model for Image Editing
## (Appendix)

**Xi Yu**[*]**, Xiang Gu**[*]**, Haozhi Liu, Jian Sun** (✉)
School of Mathematics and Statistics, Xi'an Jiaotong University, Xi'an, China
{ericayu,xianggu,liuhzh}@stu.xjtu.edu.cn {jiansun}@xjtu.edu.cn

## A   Proofs

**Lemma 1.** *Let $\beta(s) = \bar{\beta}_{\min} + s(\bar{\beta}_{\max} - \bar{\beta}_{\min})$ with $\bar{\beta}_{\max} > \bar{\beta}_{\min} > 0$. Then, for each $\lambda_k \in [0,1]$ and $t \in [0,1]$, there exists a unique time $\tau \in [0,1]$ (denoted by $\tau = \xi_k(t)$) such that $\int_0^t \beta(s)\lambda_k \mathrm{d}s = \int_0^\tau \beta(s)\mathrm{d}s$ and $\beta(t)\lambda_k \mathrm{d}t = \beta(\tau)\mathrm{d}\tau$, with the following form*

$$\xi_k(t) = \frac{-\bar{\beta}_{\min} + \sqrt{\bar{\beta}_{\min}^2 + 2(\bar{\beta}_{\max} - \bar{\beta}_{\min})\bar{\beta}_{\min}t\lambda_k + (\bar{\beta}_{\max} - \bar{\beta}_{\min})^2 t^2 \lambda_k}}{\bar{\beta}_{\max} - \bar{\beta}_{\min}}. \tag{A-1}$$

**Proof.** With the definition of $\beta(s)$, we have $\beta(s) > 0$ for any $s \in [0,1]$. Let $F(t) = \int_0^t \beta(s)\mathrm{d}s$, we have $F(t) = \bar{\beta}_{\min}t + (\bar{\beta}_{\max} - \bar{\beta}_{\min})t^2/2$ and $F(t) \in [0, (\bar{\beta}_{\max} + \bar{\beta}_{\min})/2]$ is a continuous function of $t$. Since the derivative of $F(t)$ is strictly positive, the function exhibits strict monotonic growth. Therefore, there exists the reverse function $F^{-1}$ such that $t = F^{-1}(\int_0^t \beta(s)\mathrm{d}s)$. As $0 \le \lambda_k \le 1$, we have $\int_0^t \beta(s)\lambda_k \mathrm{d}s \in [0, (\bar{\beta}_{\max} + \bar{\beta}_{\min})/2]$ for any $t$. Thus, there exists a unique time $\tau \in [0,1]$ such that $\tau = F^{-1}(\int_0^t \beta(s)\lambda_k \mathrm{d}s)$ and

$$\frac{\mathrm{d}\tau}{\mathrm{d}t} = \frac{1}{F'(\tau)} \cdot \left( \int_0^t \beta(s)\lambda_k \mathrm{d}s \right)' = \frac{\beta(t)\lambda_k}{\beta(\tau)}. \tag{A-2}$$

In Eq. (A-2), we apply the inverse function rule, and the Leibniz integral rule for differentiation under the integral sign. Therefore, we have that $\beta(t)\lambda_k \mathrm{d}t = \beta(\tau)\mathrm{d}\tau$. For a given time $t$, we now provide the specific form of $\tau$ that satisfies $\int_0^t \beta(s)\lambda_k \mathrm{d}s = \int_0^\tau \beta(s)\mathrm{d}s$. By solving the quadratic equation $\bar{\beta}_{\min}t\lambda_k + (\bar{\beta}_{\max} - \bar{\beta}_{\min})t^2\lambda_k/2 = \bar{\beta}_{\min}\tau + (\bar{\beta}_{\max} - \bar{\beta}_{\min})\tau^2/2$ with condition $\tau \in [0,1]$, we can obtain

$$\tau = \xi_k(t) = \frac{-\bar{\beta}_{\min} + \sqrt{\bar{\beta}_{\min}^2 + 2(\bar{\beta}_{\max} - \bar{\beta}_{\min})\bar{\beta}_{\min}t\lambda_k + (\bar{\beta}_{\max} - \bar{\beta}_{\min})^2 t^2 \lambda_k}}{\bar{\beta}_{\max} - \bar{\beta}_{\min}}. \tag{A-3}$$

**Theorem 1.** *For a pixel indexed by $k$, $\lambda_k \in [0,1]$, and let $\tau = \xi_k(t)$ with $\xi_k(t)$ represented in Eq. (A-3). With the same initial value $\mathbf{y}^k(0)$, we have that the transition kernel at time $t$ induced by Eq. (7) in the main paper is equal to the transition kernel at time $\tau$ induced by the following differential equation*

$$\mathrm{d}\mathbf{y}^k = -\frac{1}{2}\beta(\tau)\mathbf{y}^k \mathrm{d}\tau + \sqrt{\beta(\tau)}\mathrm{dw} \quad \text{with initial value } \mathbf{y}^k(0). \tag{A-4}$$

*The total time of noising for Eq. (A-4) is $T_k$ with $T_k = \xi_k(T)$.*

---

[*] Equal contribution.

37th Conference on Neural Information Processing Systems (NeurIPS 2023).

**Proof.** With the initial value $\mathbf{y}^k(0)$, we can compute the transition kernel using the Eqs. (5.50) and (5.51) in [1]. The transition kernel at time $t$ induced by Eq. (7) in the main paper is $\mathcal{N}\left(\mathbf{y}^k(t) \mid \mathbf{y}^k(0)e^{-\frac{1}{2}\int_0^t \beta(s)\lambda_k \mathrm{d}s}, 1 - e^{-\int_0^t \beta(s)\lambda_k \mathrm{d}s}\right)$ and the transition kernel at time $\tau$ induced by Eq. (A-4) is $\mathcal{N}\left(\mathbf{y}^k(\tau) \mid \mathbf{y}^k(0)e^{-\frac{1}{2}\int_0^\tau \beta(s)\mathrm{d}s}, 1 - e^{-\int_0^\tau \beta(s)\mathrm{d}s}\right)$.

As $\tau = \xi_k(t)$, we have $\int_0^t \beta(s)\lambda_k \mathrm{d}s = \int_0^\tau \beta(s)\mathrm{d}s$ according to Lemma 1. Thus, we have

$$
\begin{aligned}
&\mathcal{N}\left(\mathbf{y}^k(t) \mid \mathbf{y}^k(0)e^{-\frac{1}{2}\int_0^t \beta(s)\lambda_k \mathrm{d}s}, 1 - e^{-\int_0^t \beta(s)\lambda_k \mathrm{d}s}\right) \\
=&\mathcal{N}\left(\mathbf{y}^k(\tau) \mid \mathbf{y}^k(0)e^{-\frac{1}{2}\int_0^\tau \beta(s)\mathrm{d}s}, 1 - e^{-\int_0^\tau \beta(s)\mathrm{d}s}\right)
\end{aligned}
\tag{A-5}
$$

# B  Dataset Details and Additional Experimental Results

## B.1  Dataset Details

For image editing experiment on Imagen dataset, we collect synthetic images generated by Imagen [2] from https://imagen.research.google/. We use template prompts in the form of "{A photo of a} {fuzzy panda | British shorthair cat | Persian cat | Shiba Inu dog | raccoon} {wearing a cowboy hat and | wearing sunglasses and} {red shirt | black jacket} {playing a guitar | riding a bike | skateboarding} {in a garden | on a beach | on top of a mountain}". By combining these prompt templates, we generate a total of 180 images. To perform text-based image editing, we modify specific attributes within the prompts. For example, we replace "fuzzy panda" with "raccoon" in the prompt "A photo of a fuzzy panda wearing a cowboy hat and red shirt playing a guitar in a garden". This allows us to edit the images generated from the original prompts. For each image, we can conduct 10 different attribute replacements, resulting in a total of 1,800 edited images across the 180 original images.

## B.2  Qualitative Results on COCO-S and DreamBooth Datasets

We present our qualitative results on COCO-S and DreamBooth datasets in Figure 1 and Figure 2 respectively. It can be observed that our method can always edit the source image based on the target prompt, while maintaining the information irrelevant to editing unchanged, compared with the SoTA method. For example, for the image in the first column of Figure 1, DiffEdit/SDEdit/InstructPix2Pix/EDICT does not generate images that match the target text. Moreover, DiffEdit/S-DEdit/DDS/InstructPix2Pix/EDICT can not preserve the detailed background outside the bird. For the image in the first column of Figure 2, the cats in the generated images by SDEdit/DDS/EDICT do not wear a rainbow scarf. The regions below the cat in the images generated by DiffEdit/SDEd-it/DDS/InstructPix2Pix/EDICT are not similar to the corresponding region in the source image. Our method not only generates the image of a cat wearing a rainbow scarf, but also preserves the detailed background below the cat. Note that for the other columns, our method can also successfully edit based on prompt, while preserving information that is not related to editing.

## B.3  Results at the Intermediate Steps of the Forward and Reverse Process

Figure 5 illustrates the results at the intermediate steps of the forward process and reverse process, as well as the mask $\mathbf{M}$ used in the denoising process described in Algorithm 1. The masks reveal that during the translation from cat to dog, we prioritize denoising the pixels corresponding to prominent cat features, such as the eyes and nose. Subsequently, we proceed to denoise other facial regions of the cat, followed by the denoising of the background. The generated images demonstrate that we can successfully generate dog images while preserving the pose of the cat and maintaining the background unchanged.

## B.4  Computational Cost Comparision

We test the computational cost of SINE on NVIDIA Tesla V100, and the remaining methods including our NGDM on NVIDIA GeForce RTX3090. Tables 1 and 2 show the computational time and memory cost of different methods. It can be seen that our method is comparable to other methods in both computational time and memory cost since our method requires no additional training.

Table 1: Computational time and memory cost of methods with image space-based diffusion model.

| Method | SDEdit | ILVR | EGSDE | DDIB | DiffuseIT | Ours |
|---|---|---|---|---|---|---|
| Time per iteration (s) $\downarrow$ | **18** | 44 | 62 | 210 | 48 | 42 |
| Memory(GB) $\downarrow$ | 3.3 | **2.8** | 4.5 | 3.8 | 16.6 | 7.4 |

Table 2: Computational time and memory cost of methods with latent space-based diffusion model.

| Method | SDEdit | DiffEdit | SINE | DDS | InstructPix2Pix | EDICT | Ours |
|---|---|---|---|---|---|---|---|
| Time per iteration (s) $\downarrow$ | **3** | 9 | 3480 | 46 | 12 | 648 | 6 |
| Memory(GB) $\downarrow$ | 10.0 | **6.7** | 28.0 | 16.7 | 18.0 | 13.8 | **6.7** |

## B.5 More Tasks

We conduct experiments on two additional tasks to explore the potential of our approach, including style transfer and gender transformation. The visual results are shown in Figures 3 and 4. The style transfer task aims to transform the image into another style without changing the structure. For example, from Figure 3, we can see that our method can transform a "real dog" into a "sculptural dog" without changing the structure of the dog. From Figure 4, our method can turn males into females while keeping the structure of the face unchanged.

## B.6 More Qualitative Results

In this section, we show more qualitative results in Figures 6, 7, and 8 using the default hyper-parameters. Besides, in Figures 9 and 10 we provide the visualization results of three methods of DDS [3], InstructPix2Pix [4], and EDICT [5], in the examples in Figures 3 and 4 of the main paper.

## B.7 More Examples Generated Using Hard Weighting Matrix

In this section, we show more examples generated using hard weighting matrix in Figure 11.

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

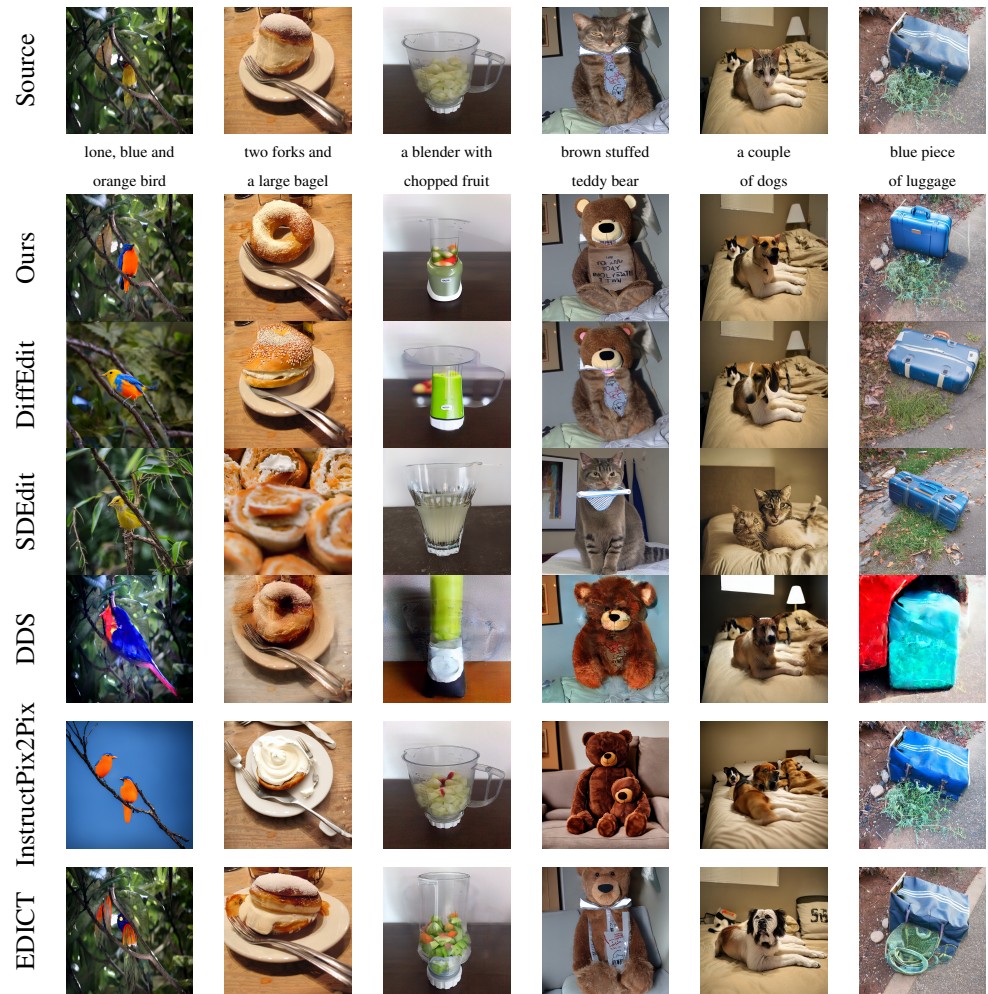

Figure 1: Qualitative comparison on COCO-S dataset. The text below the source image represents the simplified target prompt.

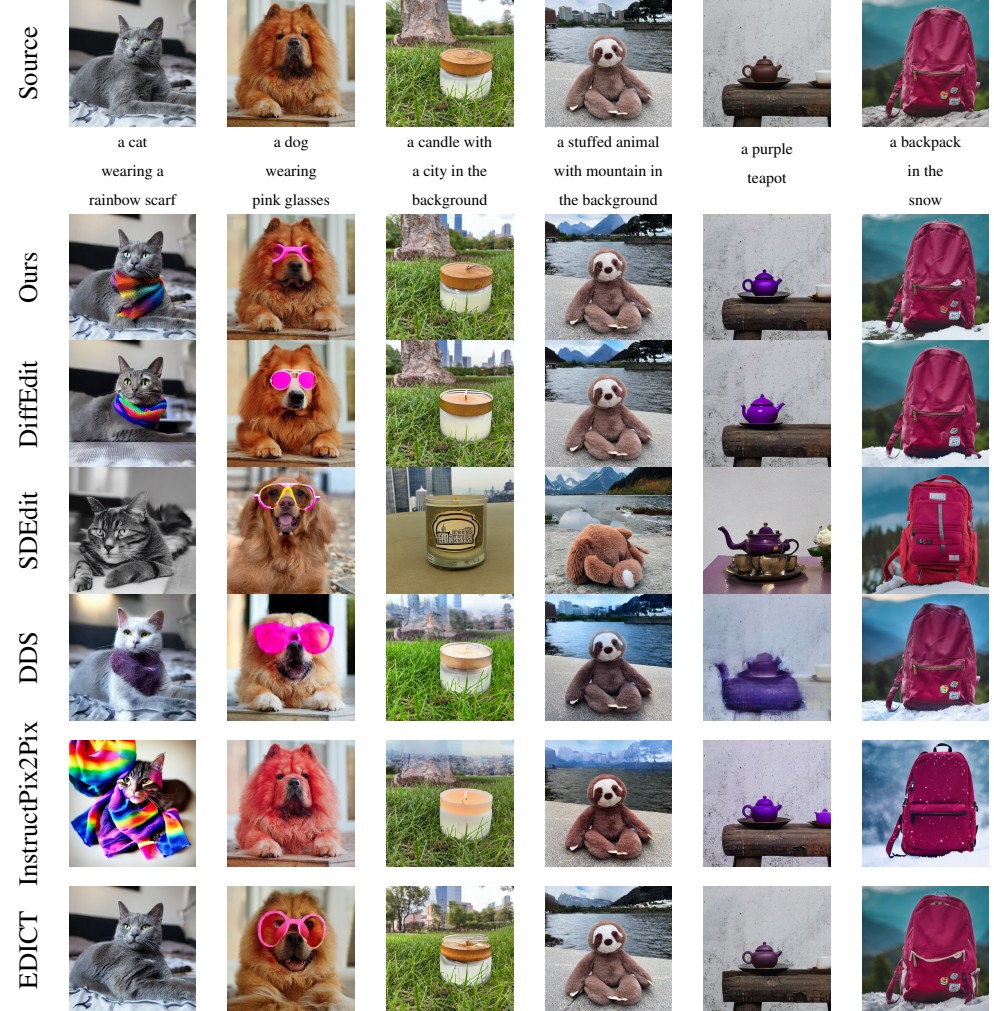

Figure 2: Qualitative comparison on Dreambooth dataset. The text below the source image represents the target prompt.

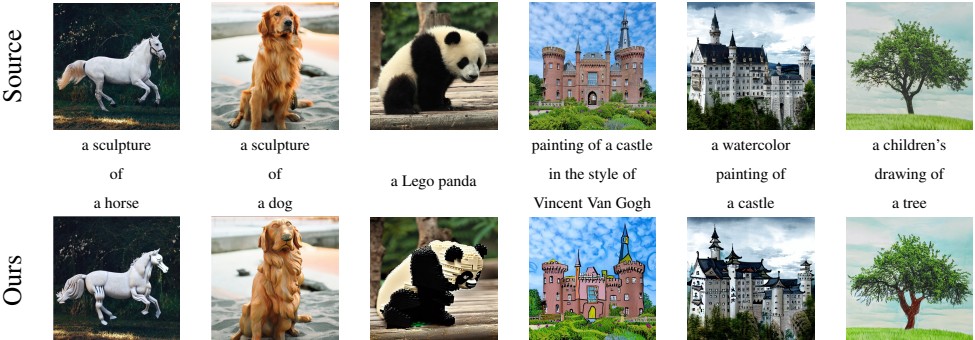

Figure 3: Examples of style transfer. The text below the source image represents the target prompt

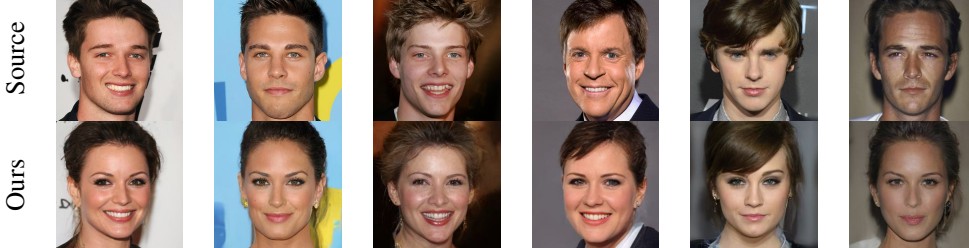

Figure 4: Examples of gender transformation.

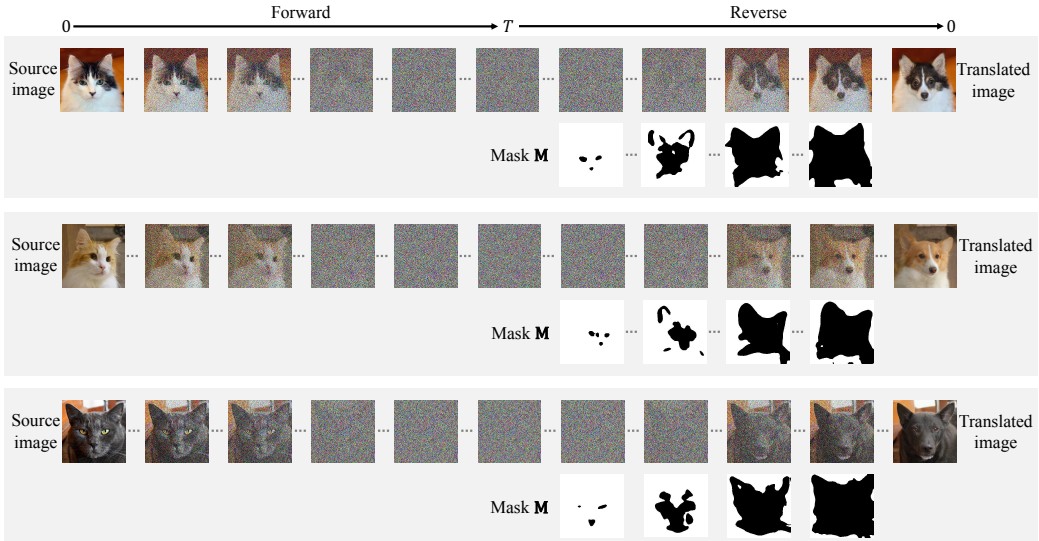

Figure 5: Results at the intermediate steps of the forward process and reverse process. We present the results at the intermediate steps of the forward and reverse process, along with the mask M at the intermediate steps during the denoising, which is described in the sixth line of Algorithm 1 in the main paper. The mask image at each time step indicates the regions in which the denoising has already started and the regions where it has not yet started. The black regions in the mask image indicate the pixels that have undergone the denoising process, while the white regions represent the pixels that have not yet been denoised.

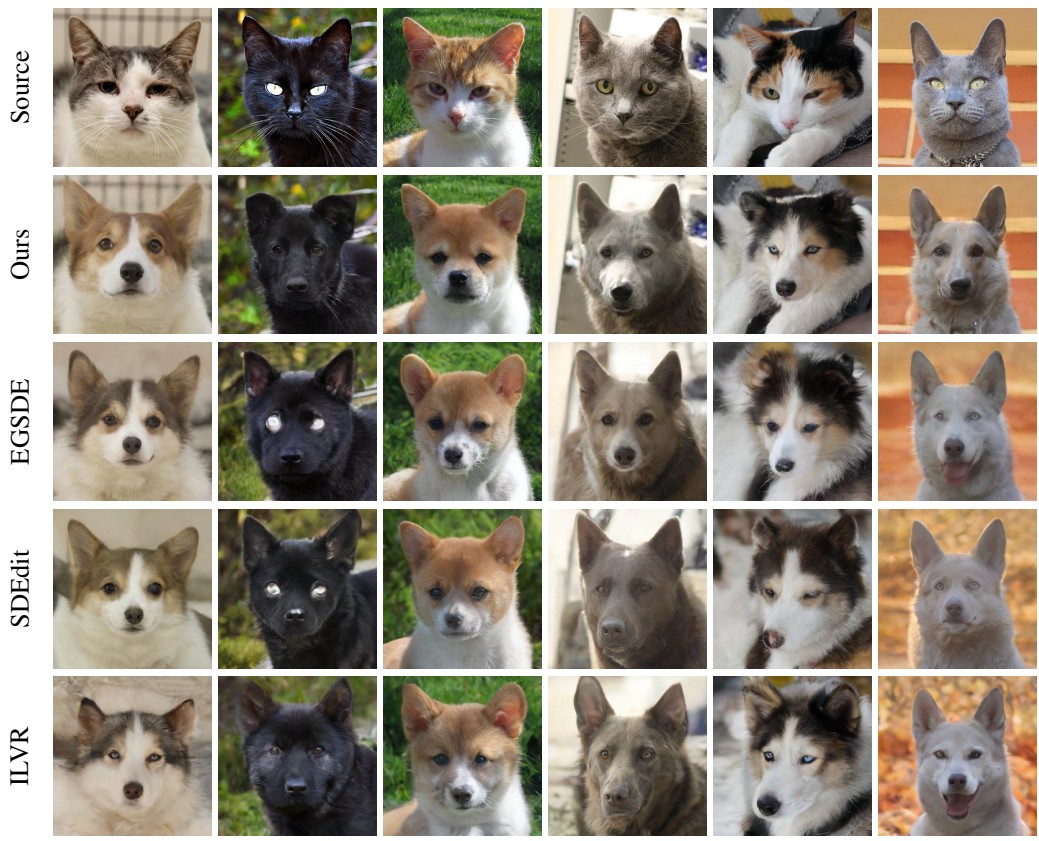

Figure 6: Qualitative comparison on AFHQ dataset.

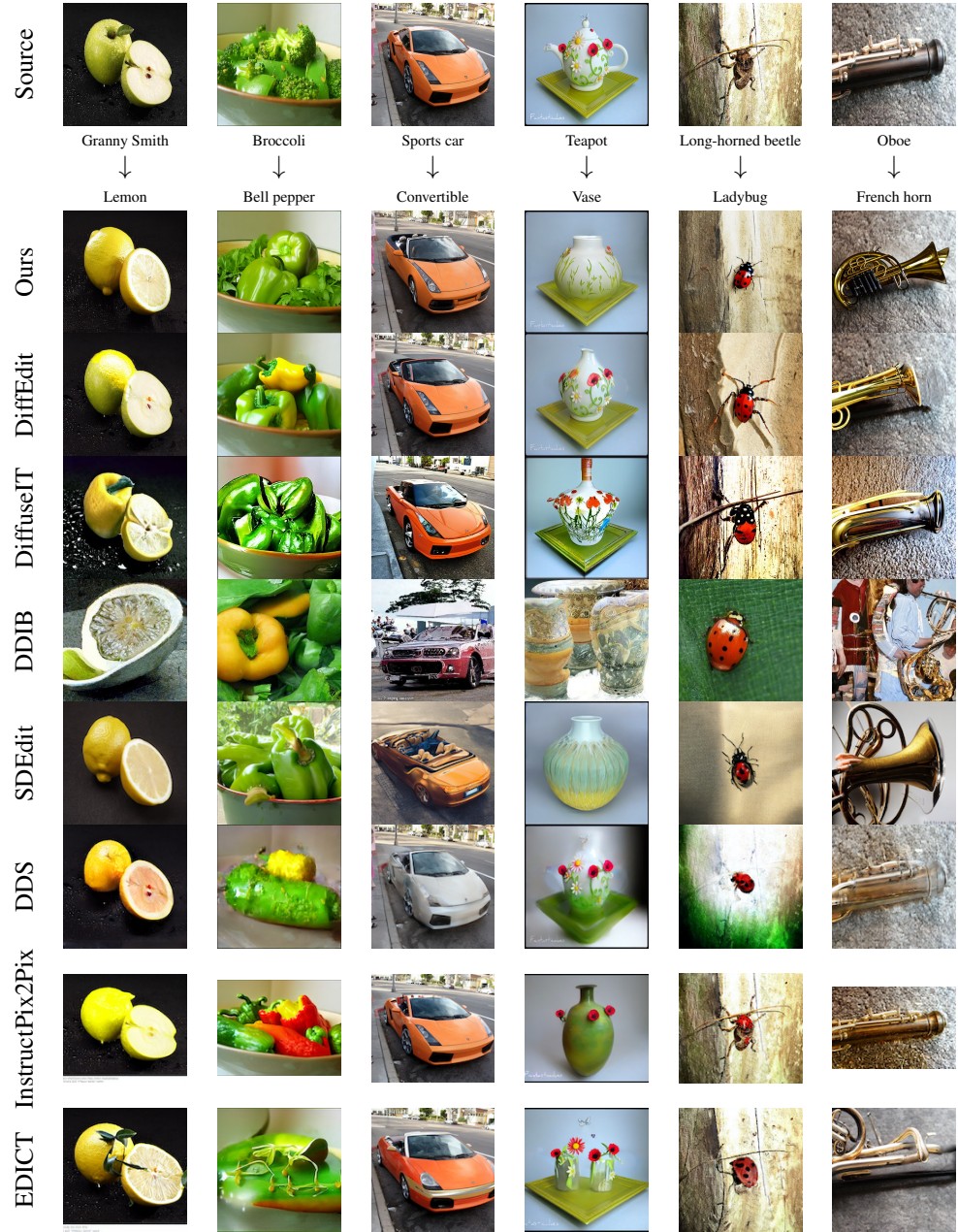

Figure 7: Qualitative comparison on ImageNet dataset.

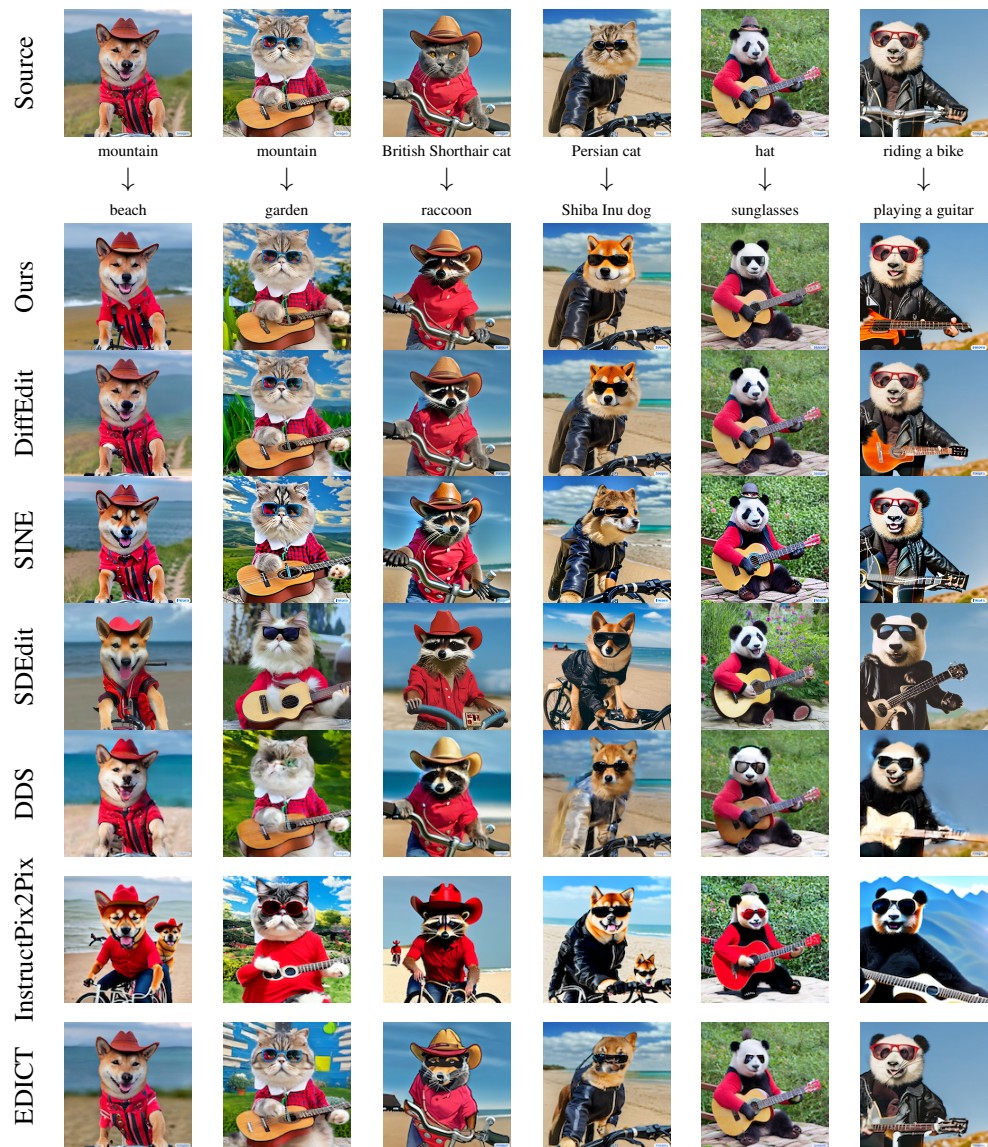

Figure 8: Qualitative comparison on Imagen dataset.

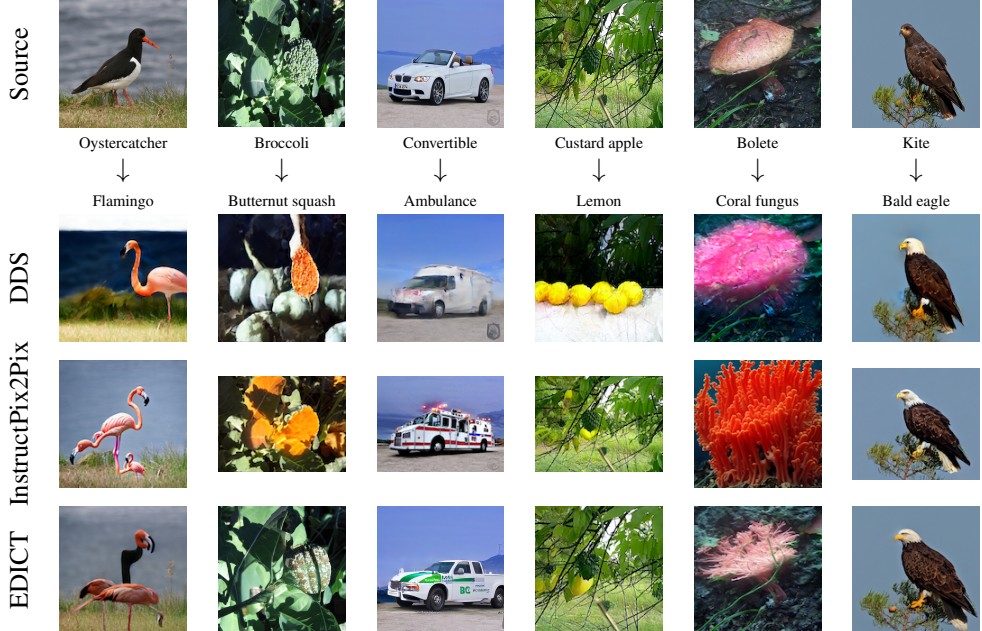

Figure 9: Visualization results of DDS, InstructPix2Pix, and EDICT on ImageNet dataset.

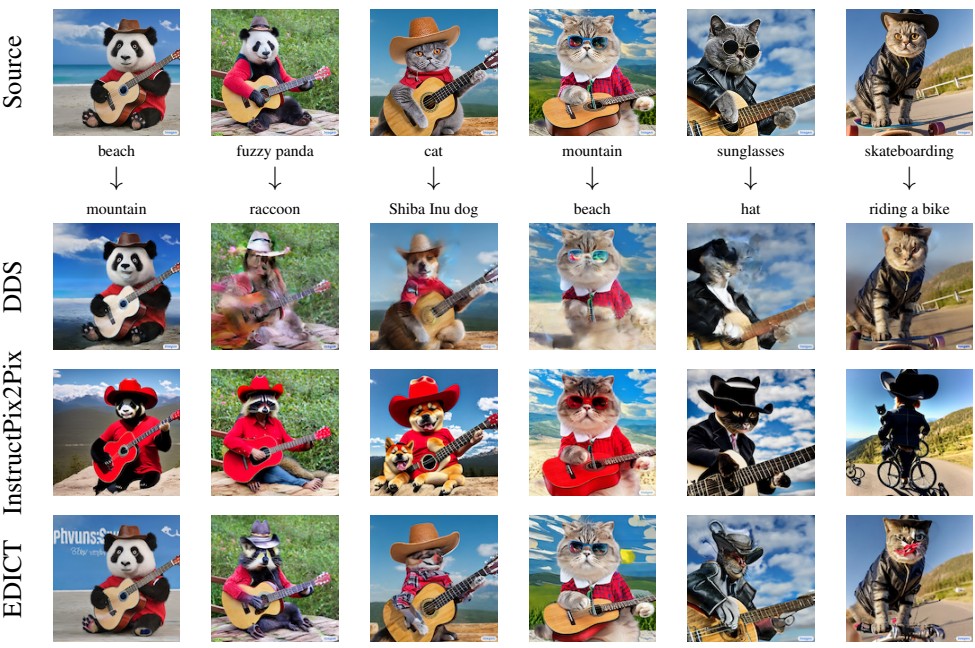

Figure 10: Visualization results of DDS, InstructPix2Pix, and EDICT on Imagen dataset.

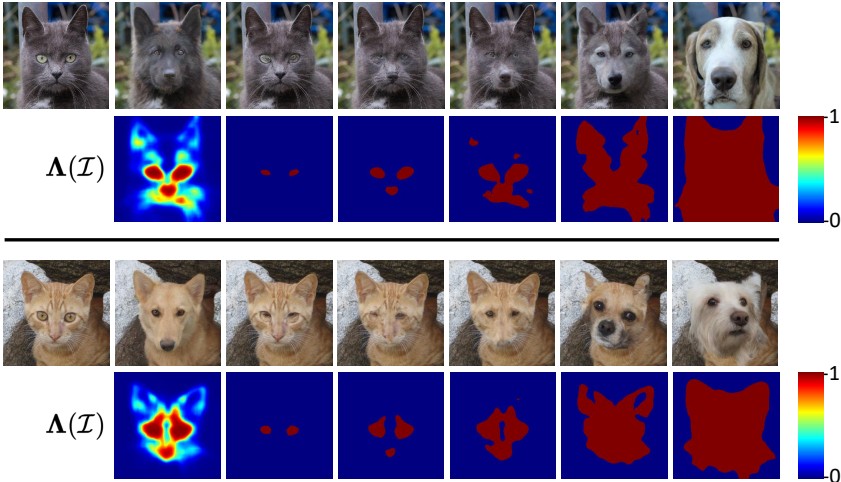

Figure 11: Edited images and heatmaps with soft and hard weighting matrix. The images in the second column represent the results generated by our method and the heatmap below the image depicts the weighting matrix $\mathbf{\Lambda}(\mathcal{I})$ defined in Section 3.1 in the paper. The images in columns 3-7 represent the results generated using the hard weighting matrix with threshold value in $\{0.1, 0.3, 0.5, 0.7, 0.9\}$. The heatmaps below the images represent the binary hard weighting matrix.