# OpenReview forum: "Constructing Non-isotropic Gaussian Diffusion Model Using Isotropic Gaussian Diffusion Model for Image Editing"
_NeurIPS.cc/2023/Conference — NeurIPS 2023 poster_

### Official Review · Reviewer_LtuA · 2023-07-01

**Soundness:** 3 good
**Presentation:** 2 fair
**Contribution:** 2 fair
**Rating:** 4
**Confidence:** 4

**Summary:**

Gaussian diffusion is a common concept used in image processing, physics, and machine learning. It refers to a process where values (like the pixels in an image) are "smoothed out" or diffused according to a Gaussian function (also known as a normal distribution or bell curve), which provides a natural and mathematically convenient model for this diffusion process.  In the context of Gaussian diffusion, the term "isotropic" means that the diffusion is uniform in all directions. In contrast, "non-isotropic" Gaussian diffusion means that the diffusion is not uniform in all directions. The amount a pixel gets diffused might depend on its direction.

The authors propose a method to use a pre-trained Isotropic Gaussian Diffusion Model (IGDM) for sampling in the context of a Non-isotropic Gaussian Diffusion Model (NGDM). In the first step, they define the NGDM with added independent non-isotropic Gaussian noise. This suggests that they are introducing a type of Gaussian noise that's directionally dependent, in line with the concept of non-isotropic diffusion. Then, they detail how to implement the NGDM using the pre-trained IGDM. This process involves 'rectifying' the spatially different times of the noise and denoise procedures in the NGDM. It seems they're addressing the inherent differences between isotropic and non-isotropic models and adjusting the processes accordingly. Finally, they present a data sampling algorithm for the proposed NGDM that uses the pre-trained IGDM. This suggests that they're leveraging the pre-trained model's capabilities in the new context of non-isotropic diffusion.



**Strengths:**

1. Overall, they're taking an existing model trained on isotropic diffusion, adapting it for non-isotropic diffusion, and presenting a new algorithm for sampling data within this context.

2. The authors show through experiments that their proposed method to be competitive.

**Weaknesses:**

1. The authors discussed the settings of the parameters \alpha and \beta but I do not see a strong link to the quality of the generated images.

2. From the generated images shown in Figures 2-4, 6, I cannot really say that the proposed method outperforms the other consistently.  There is simply no "wow" effect.  Figure 6 shows that the color of the head also changed, not described in "black leather jacket".



**Questions:**

1. Please compare computational cost of all involved methods resented in Section 4.
2. Please show a couple of examples where NGDM performs worse, and give the reasons to explain why.

**Limitations:**

It is simply impossible to thoroughly "prove" a method is better by just displaying some selected results.  A much more robust and comprehensive empirical study followed by why the proposed NGDM works better would be helpful.

---

> ### Author Rebuttal · Authors · 2023-08-10
>
> **Q1: The link of $a$ and $b$ to the quality of the generated images.**
>
> We present the visual results when the hyper-parameters $a$ and $b$ vary respectively in Figure 6 of the main paper. Keeping $b$ constant, when $a$ is small, all element values of the weighting matrix obtained after Sigmoid transformation are relatively small. At this point, the noise variance on each pixel of the image is small, resulting in that the generated image is similar to the source image. However, the generated image does not align well with the target prompt. When $a$ gets larger, the generated image begins to align with the target prompt. Keeping $a$ constant, $b$ does the opposite. When $b$ is small, the generated image does not preserve well the source image information. When $b$ gets larger, the generated image will preserve more source image information.
>
> **Q2: The proposed method does not outperform the other consistently. Figure 6 shows that the color of the head also changed, not described in "black leather jacket".**
>
> We focus on controllable image editing that edit image based on target prompt while minimally modifying the source image. Compared with other methods, our method can better preserve the background, pose, etc. For example, for images with complex backgrounds in columns 2-5 of Figure 2, our method can accurately keep the background of the source image unchanged while translating cats into dogs. Other methods either blur the background or fail to maintain the source image background. In Figure 3, the source images in columns 2 and 4 have detailed backgrounds, and our method can preserve these details to the greatest extent while editing images.
>
> As we analyzed above and the results shown in Figure 6 of the main paper, when $a$ is 20 or $b$ is 0, the source image information cannot be well maintained, resulting in changes in the head's color. We can achieve a balance between the two by choosing appropriate parameters. When $a=10.0$ and $b=5.0$, our method can preserve the head color described in "black leather jacket".
>
> **Q3: Computational cost comparison.**
>
> Tables r6-1 and r6-2 show the computational efficiency comparison. Our method has relatively small inference time and requires no additional training.
>
> Table r6-1: Computational time and memory cost of methods with image space-based diffusion model.
>
> |Method|SDEdit|ILVR|EGSDE|DDIB|DiffuseIT|NGDM (Ours)|
> |:-:|:-:|:-:|:-:|:-:|:-:|:-:|
> |Time per iteration (s)$\downarrow$|**18**|44|62|210|48|42|
> |Memory(GB)$\downarrow$|3.3|**2.8**|4.5|3.8|16.6|7.4|
>
> Table r6-2: Computational time and memory cost of methods with latent space-based diffusion model.
>
> |Method|SDEdit|DiffEdit|SINE|DDS|InstructPix2Pix|EDICT|NGDM (Ours)|
> |:-:|:-:|:-:|:-:|:-:|:-:|:-:|:-:|
> |Time per iteration (s)$\downarrow$|**3**|9|3480|46|12|648|6|
> |Memory(GB)$\downarrow$|10.0|**6.7**|28.0|16.7|18.0|13.8|**6.7**|
>
> **Q4: A couple of examples where NGDM performs worse and reasons for explaining why.**
>
> We show a few failure examples in Figure 6 of the Appendix. We think that the failure of the image in the first column may be due to the artifacts generated by the underlying model we depend on, anf the failures of images in columns 2 and 3 may be due to that the computed weighting matrix is not accurate.
>
> **Q5: A much more robust and comprehensive empirical study followed by why the proposed NGDM works better.**
>
> We achieve controllable image editing by adding a corresponding degree of noise to each pixel to the extent that it needs to be edited. Our method can effectively preserve the original content by adding noise with small variances to the regions irrelevant to the editing task. Compared with mask-guided image editing, such as DiffEdit, our method can avoid the edge artifact problem caused by the mask.
>
> We add more experimental results to justify the effectiveness of our proposed NGDM, including the comparison results on more datasets and more SoTA methods, the results of user study, and the results of computation efficiency.
>
> We add two natural datasets (COCO-S and DreamBooth Dataset [Ruiz N, et al., CVPR2023]), and three SoTA methods DDS [Hertz A, et al., ], InstructPix2Pix [Brooks T, et al., CVPR2023] and EDICT [Wallace B, et al., CVPR2023] for comparison. The quantitative results shown in Table r6-3 indicate that our method outperforms competing methods by achieving a better trade-off between CLIPScore and LPIPS value.
>
> Table r6-3: Quantitative comparison on COCO-S dataset and Dreambooth dataset.
>
> |Method|SDEdit|DiffEdit|DDS|EDICT|InstructPix2Pix|Ours ($a$=10.0,$b$=5.0)|Ours ($a$=10.0,$b$=6.0)|
> |:-:|:-:|:-:|:-:|:-:|:-:|:-:|:-:|
> |CLIPScore$\uparrow$ (COCO-S dataset)|28.58|30.31|30.48|30.77|31.28|**31.75**|31.45|
> |LPIPS$\downarrow$ (COCO-S dataset)|55.76|29.60|31.53|24.00|36.55|28.80|**23.43**|
> |CLIPScore$\uparrow$ (Dreambooth dataset)|19.64|19.88|**20.17**|19.70|19.64|19.81|19.75|
> |LPIPS$\downarrow$ (Dreambooth dataset)|59.82|27.47|33.26|22.52|40.50|24.86|**19.95**|
>
> We conduct user study by inviting 40 participants and providing each of them with 30 randomly selected source images where the corresponding generated results of different methods are displayed randomly. Participants were asked to choose the image that better applies the requested edit while preserving most of the original image details. The percentage of votes for each method is shown in Tables r6-4 and r6-5, which demonstrate that the participants exhibit a strong preference for our method.
>
> Table r6-4: User study results on Cat $\rightarrow$ Dog task.
>
> |ILVR|SDEdit|EGSDE|Ours|
> |:-:|:-:|:-:|:-:|
> |11.5%|10.5%|12.5%|**65.5%**|
>
> Table r6-5: User study results on the remaining tasks.
>
> |SDEdit|DiffEdit|DDS|EDICT|InstructPix2Pix|Ours|
> |:-:|:-:|:-:|:-:|:-:|:-:|
> |4.5%|10.0%|3.0%|4.5%|6.0%|**72.0%**|

---

> > ### Author Response · Authors · 2023-08-17
> > **Further clarification on qualitative comparisons in Figures, effects of $a$ and $b$, more experiments and analysis.**
> >
> > Dear reviewer, thanks for your comments and questions. Besides the above rebuttal, in the author-reviewer discussion phase, we would like to further clarify as follows.
> >
> > (1) **Additional clarification on qualitative comparisons in Figures and user study.** Beside the rebuttal, we additionally clarified the qualitative comparisons and user studies in the official comment with the title "To ACs and Reviewers: further clarification on the qualitative comparison with other methods and on the user study in the responses", following the "Author Rebuttal by Authors". Please refer to them for detailed clarifications.
> >
> > (2) **The effects of $a$ and $b$ on the quality of generated images.** Besides the responses to Q1, we have also reported the metrics of FID and SSIM (on Cat $\rightarrow$ Dog translation task) in Table 2 of the paper, by fixing one and changing the other. Please refer to the responses to Q1 and Lines 193-200 in the paper for the discussions.
> >
> > (3) **About the selection of the displayed examples.** We clarify that the displayed examples in Figures 2-4 are randomly selected from each dataset for fair comparisons. Note that due to space limit, we can only show a few examples, but overall these conclusions are the same in the other examples of these images in the datasets.
> >
> > (4) **More comprehensive experimental study.** In the rebuttal, we additionally conduct experiments on two natural datasets (COCO-S and DreamBooth Dataset) and additionally compare with three SoTA methods DDS, InstructPix2Pix, and EDICT. The quantitative results are shown in Table r6-3, and the qualitative results are shown in Figure r1 in the uploaded one-page pdf in the Author Rebuttal with the title "Author Rebuttal by Authors". Additionally, we conduct user studies in the rebuttal, of which the results are reported in Tables r6-4 and r6-5. Please refer to them.
> >
> > (5) **Why NGDM works better.** Our method works better mainly because it is based on the proposed non-isotropic Gaussian diffusion model that takes different noise variances for different pixels in the Gaussian diffusion model. The input image is edited based on adding non-isotropic noises and then denoised by Algorithm 1 in the paper to generate the edited image, using the pre-trained isotropic Gaussian diffusion model. The pixel's noise variance is computed based on Eq. (8), according to the pixel's relevance to the editing task using the soft weighting matrix $\lambda(I)$. The pixels with higher noise variance will be edited more heavily than the pixels with smaller noise variance, to ensure that the input image is correctly edited while preserving the image regions unrelated to the editing task unchanged. The experiments and qualitative comparisons have demonstrated the advantage of our method.

---

> ### Comment · Area_Chair_6fEM · 2023-08-18
>
> Reviewer LtuA: Please respond to the author's rebuttal ASAP.

---

### Official Review · Reviewer_ev4D · 2023-07-03

**Soundness:** 2 fair
**Presentation:** 2 fair
**Contribution:** 2 fair
**Rating:** 5
**Confidence:** 1

**Summary:**

This paper presents a novel Non-isotropic Gaussian Diffusion Model (NGDM) for image-to-image translation and image editing tasks. The central idea of NGDM is to add independent Gaussian noises with different variances to different pixels, thus achieving controllable image translation and editing based on the amount of noise variance added to each pixel. Unlike prior models, the NGDM doesn't need specific training; instead, it rectifies into an isotropic Gaussian diffusion model where different pixels have varying total forward diffusion time.

To generate images from the diffused ones, the authors propose a sampling method that initiates denoising at different times for different pixels using a pre-trained isotropic Gaussian diffusion model. This process allows for the preservation of certain parts of the image while editing or translating others, thereby providing flexibility in performing such tasks.

The paper demonstrates the effectiveness of NGDM through experiments on three datasets encompassing real and synthetic images. The results show that NGDM outperforms state-of-the-art score-based diffusion models (SBDMs) in terms of FID and SSIM metrics for Cat → Dog image translation task. Similar superior performance was also observed in text-guided image translation and image editing tasks.

In summary, the main contributions of the paper are:
1. The proposal of a novel Non-isotropic Gaussian Diffusion Model (NGDM) for image-to-image translation and image editing.
2. Demonstration of how to rectify the NGDM into an isotropic Gaussian diffusion model, which allows leveraging a pre-trained isotropic model for denoising and image generation.
3. A new sampling method for generating images by starting denoising at different times for different pixels.
4. Validation of the proposed model against state-of-the-art SBDMs across several tasks and datasets, showing improved performance.
5. Exploration of the trade-off between image fidelity and alignment with desired translation/editing targets, providing an avenue for controlling this balance with varying hyperparameters.

**Strengths:**

**Originality:** The proposed Non-isotropic Gaussian Diffusion Model (NGDM) demonstrates a high level of originality. It creatively deviates from the standard practice of isotropic diffusion by adding independent Gaussian noises with different variances to different pixels, enabling more controllable image translation and editing. This concept is novel and adds an interesting dimension to image generation models.

**Quality:** The paper is of high quality. The authors provide a comprehensive explanation of the model, including detailed descriptions of the methodology. They describe the process of rectifying NGDM into an isotropic Gaussian diffusion model, which allows leveraging a pre-trained model for denoising and image generation. The experimental results on multiple datasets convincingly support their claims, showing that their method outperforms state-of-the-art score-based diffusion models in several tasks.

**Clarity:** The paper is well-structured and clear. The authors do an excellent job explaining complex concepts in an understandable manner, making the paper accessible to readers with varying levels of familiarity with the topic. Diagrams and visual aids would further enhance the clarity of the paper.

**Significance:** This work holds significant potential for advancing the field of image-to-image translation and image editing. The method provides a new way to control the translation/editing process by varying the noise added to different pixels, which could be invaluable in numerous applications. Moreover, this model's superior performance across multiple tasks and datasets shows promise for practical use in real-world scenarios. By demonstrating how to leverage existing isotropic Gaussian diffusion models within the NGDM framework, the authors also open up possibilities for future research in this area.

**Weaknesses:**

While the paper presents a novel and promising approach in the Non-isotropic Gaussian Diffusion Model (NGDM) for image-to-image translation and image editing tasks, there are several areas that could be improved:

1. **Contextual Understanding:** The authors might consider providing more information on how their work compares to other existing methods. While they mention a few specific models and techniques, a more extensive literature review and comparison would be beneficial. This would allow readers to better understand the novelty and advantages of NGDM.

2. **Experimentation:** The experiments conducted show promising results; however, the testing could be expanded. The authors only use three datasets for validation. More diverse datasets could help provide a more comprehensive performance evaluation of the method across different scenarios. Also, it would be helpful if the authors could include comparative visual results along with the quantitative metrics.

3. **Model Interpretability:** While the model's performance is impressive, its interpretability seems not addressed. It might be challenging to understand why specific noise variances are assigned to certain pixels during the diffusion process. Providing some insights or analysis into this aspect may make the model more understandable.

4. **Potential Limitations and Pitfalls:** The paper lacks a discussion about potential limitations and considerations when using the proposed model. Addressing possible issues such as computational cost, scalability, and any conditions under which the model might fail or perform suboptimally would provide a more balanced and realistic view of the proposed method.

5. **Societal Impact:** As with any machine learning model, it's crucial to consider the ethical implications and potential misuse cases. Given the model's capacity for image manipulation, discussions around data privacy and consent, as well as potential ways the technology could be misused, should be included.

These suggestions should help to strengthen the paper and broaden the understanding and applicability of the proposed NGDM.


**Questions:**

Here are a few questions and suggestions that would benefit from further clarification by the authors:

1. **Noise Variance Distribution:** How is the variance of the Gaussian noise chosen for each pixel? Is there any strategy or algorithmic process behind this, or is it random? Knowing more about this could help understand how the NGDM manages to preserve certain parts of an image while translating/editing others.

2. **Comparison with Other Models:** Could you provide a more detailed comparison between NGDM and other state-of-the-art models? It would be beneficial to include both qualitative (visual comparisons) and quantitative (additional metrics) results where possible.

3. **Performance on Different Scenarios:** How does the model perform when applied to different tasks, especially those that have not been covered in the experimental section? Understanding its versatility and limitations across various scenarios will paint a fuller picture of NGDM's applicability.

4. **Computational Efficiency:** Could you elaborate on the computational efficiency of your model? Specifically, how does the time complexity of the proposed method compare to other isotropic diffusion models? Does adding noise with different variances to different pixels significantly increase computational cost?

5. **Ethical Considerations:** As your model allows substantial manipulation of images, what ethical considerations should be taken into account? Discussions around potential misuse, data privacy, and consent would add value.

6. **Potential Improvements:** Lastly, can you suggest potential avenues for further improving the performance of NGDM? This might include parameter tuning, integrating with other models, or even extending the model to other domains beyond image translation and editing.

Looking forward to your responses to these queries and suggestions, which I believe will provide a clearer understanding of your work and its implications.

**Limitations:**

Based on the provided information, it appears that the authors have not thoroughly addressed the potential limitations and negative societal impacts of their work.

Here are a few suggestions for improvement:

**Limitations:**

1. **Computational Efficiency:** The authors should address the computational efficiency of their proposed model. Adding noise with different variances to different pixels might increase computational cost, which could be a limitation in some scenarios.

2. **Generalizability:** The paper does not discuss the performance of the model when applied to other tasks or datasets beyond those tested. Addressing this would give a clearer picture of the model's versatility and robustness.

3. **Model Interpretability:** The interpretability of the NGDM also seems unclear. Understanding why specific noise variances are assigned to certain pixels during the diffusion process could be complex. Discussing these aspects would help readers better understand and apply the model.

**Societal Impacts:**

1. **Ethical Considerations:** As the proposed model allows substantial manipulation of images, it is important to consider the ethical implications. The authors should provide a discussion about potential misuse, data privacy, and consent.

2. **Potential Misuse:** With advanced image editing and translation capabilities, there may be potential for misuse of the technology, such as deepfake creation or unauthorized alteration of images. The authors should address these concerns and possible measures to prevent misuse.

Discussing these potential limitations and societal impacts would create a more balanced view of NGDM and help prepare users for any challenges they might face when applying the model.

---

> ### Author Rebuttal · Authors · 2023-08-10
>
> **Q1: Comparison with more methods and experiments on more datasets.**
>
> For Cat $\rightarrow$ Dog translation task, we add SDDM [Sun S, et al., ICML2023] for comparison. SDDM decomposes the score function into an image “denoising” part and a content “refinement” part for translation. Differently, we perform translation by adding independent noise with different variances to different pixels. Results in Table r5-1 show that our method achieves the best results among the compared methods.
>
> Table r5-1: Quantitative comparison on Cat $\rightarrow$ Dog translation task.
>
> |Method|ILVR|SDEdit|EGSDE|SDDM|NGDM (Ours)|
> |:-:|:-:|:-:|:-:|:-:|:-:|
> |FID($\downarrow$)|74.37±1.55|74.17±1.01|65.82±0.77|62.29±0.63|**61.39±0.27**|
> |SSIM($\uparrow$)|0.363±0.001|0.423±0.001|0.415±0.001|0.422±0.001|**0.478±0.001**|
>
> For image editing task, we add DDS [Hertz A, et al., arXiv:2304.07090, 2023], InstructPix2Pix [Brooks T, et al., CVPR2023] and EDICT [Wallace B, et al., CVPR2023] for comparison. DDS utilizes delta scoring to provide effective gradients for editing. InstructPix2Pix trains a conditional diffusion model for editing that enables zero-shot generalization. EDICT proposes exact inversion of real and generated images. Differently, we edit the image by adding independent noise with different variances to different pixels.
>
> We additionally consider two natural datasets (COCO-S and DreamBooth Dataset [Ruiz N, et al., CVPR2023]). The qualitative results are shown in Figure r1 in the uploaded one-page pdf. The quantitative results shown in Table r5-2 indicate that our method outperforms competing methods by achieving a better trade-off between CLIPScore and LPIPS value.
>
> Table r5-2: Quantitative comparison with different methods on four datasets.
>
> |Method|SDEdit|DiffEdit|DDS|EDICT|InstructPix2Pix|Ours ($a$=10.0,$b$=5.0)|Ours ($a$=10.0,$b$=6.0)|
> |:-:|:-:|:-:|:-:|:-:|:-:|:-:|:-:|
> |CLIPScore$\uparrow$ (Imagenet)|30.39|28.98|28.13|27.64|29.28|**30.66**|29.61|
> |LPIPS$\downarrow$ (Imagenet)|58.73|32.40|43.57|44.93|41.41|37.10|**31.32**|
> |CLIPScore$\uparrow$ (Imagenen)|35.81|35.19|34.69|34.75|**36.06**|35.66|35.53|
> |LPIPS$\downarrow$ (Imagenen)|50.84|21.81|31.38|**19.31**|46.07|24.29|20.28|
> |CLIPScore$\uparrow$ (COCO-S)|28.58|30.31|30.48|30.77|31.28|**31.75**|31.45|
> |LPIPS$\downarrow$ (COCO-S)|55.76|29.60|31.53|24.00|36.55|28.80|**23.43**|
> |CLIPScore$\uparrow$ (Dreambooth)|19.64|19.88|**20.17**|19.70|19.64|19.81|19.75|
> |LPIPS$\downarrow$ (Dreambooth)|59.82|27.47|33.26|22.52|40.50|24.86|**19.95**|
>
> **Q2: Model interpretability.**
>
> Image editing task aims to modify the source image under target guidance while leaving the regions that are unrelated to the editing task unchanged.
>
> It is empirically known that the diffusion model can generate more diverse novel content if adding noise with larger variance to the image, and preserve the content if adding smaller variance noise. Motivated by this, we achieve controllable editing by adding different variances of noise to different pixels.
>
> As illustrated in the lower part of Figure 1, with $\mathbf{\Lambda}(\mathcal{I})$ computed on the source image by method DiffEdit in Section 3.4, the nose region is given with large variance noise for translating cat to dog, while the background is preserved by adding small variance noise.
>
> **Q3: Potential limitations and improvements.**
>
> A limitation of our method could be that incorrect weighting matrix may lead to the failure of the method. Moreover, our method relies on a pre-trained diffusion model. Artifacts are produced when the edit involves generation failure cases of the underlying model. We will add the limitations in the paper.
>
> In future work, we will design a better way to calculate the weighting matrix more precisely and efficiently.
>
> **Q4: Societal impact, ethical considerations and potential misuse.**
>
> In our experiments, all the considered datasets are open-sourced and publicly available. Our work aims to manipulate images with minimum effort. However, this method might be misused by faking images. We will take care to exploit the method to avoid the potential negative social impact and we will help research in identifying and preventing malicious editing. To mitigate potential misuse, we will release our code under a license focused on ethical and legal use, stating explicitly that illegal and unethical use will not be allowed.
>
> **Q5: Noise Variance Distribution.**
>
> We described in Section 3.4 of the main text about how to choose the variance of the Gaussian noise for each pixel.
>
> **Q6: Performance on different scenarios.**
>
> We perform two tasks that were not covered in the experimental section, including local style transfer and gender transformation. The local style transfer task aims to transform the specified object in the image into another style while preserving the content of the rest of the region. Gender transformation aims to transform male into female. Due to space limitations, we show a few examples in Figure r2 of the uploaded one-page pdf, and we will show more examples in Appendix B in the revised version.
>
> **Q7: Computational efficiency.**
>
> Tables r5-4 and r5-5 show the computational time and memory cost of different methods. Our method is comparable to the other methods in both computational time and memory cost.
>
> Table r5-4: Computation time and memory of methods with image space-based diffusion model.
>
> |Method|SDEdit|ILVR|EGSDE|DDIB|DiffuseIT|NGDM (Ours)|
> |:-:|:-:|:-:|:-:|:-:|:-:|:-:|
> |Time per iteration (s)$\downarrow$|**18**|44|62|210|48|42|
> |Memory(GB)$\downarrow$|3.3|**2.8**|4.5|3.8|16.6|7.4|
>
> Table r5-5: Computation time and memory of methods with latent space-based diffusion model.
>
> |Method|SDEdit|DiffEdit|SINE|DDS|InstructPix2Pix|EDICT|NGDM (Ours)|
> |:-:|:-:|:-:|:-:|:-:|:-:|:-:|:-:|
> |Time per iteration (s)$\downarrow$|**3**|9|3480|46|12|648|6|
> |Memory(GB)$\downarrow$|10.0|**6.7**|28.0|16.7|18.0|13.8|**6.7**|

---

> > ### Comment · Reviewer_ev4D · 2023-08-18
> >
> > I acknowledge I have read the rebuttal.

---

> > > ### Author Response · Authors · 2023-08-18
> > > **Thank Reviewer ev4D for the comments.**
> > >
> > > Thanks, and we will carefully revise the paper according to these questions and comments in the reviews.

---

> > > ### Comment · Area_Chair_6fEM · 2023-08-18
> > >
> > > Reviewer rj2K: We need more than acknowledgement that you've read the rebuttal. Please state your assessment of the rebuttal and whether this changes your opinion of the paper.

---

### Official Review · Reviewer_rj2K · 2023-07-04

**Soundness:** 3 good
**Presentation:** 3 good
**Contribution:** 2 fair
**Rating:** 5
**Confidence:** 3

**Summary:**

The proposed model use differentiated reverse sampling strategy for image editing and translation.

**Strengths:**

The proposed model use differentiated reverse sampling strategy for image editing and translation.

**Weaknesses:**

1. Please include the user study results on evaluating the natural image editing for qualitative evaluation.

2. The method lacks novelty. Using different starting point with masked and non-masked region is just a variation of DiffEdit approach. Please elaborate the difference between baseline models.

**Questions:**

See the weakness part.

---

> ### Author Rebuttal · Authors · 2023-08-10
>
> **Q1: Please include the user study results on evaluating the natural image editing for qualitative evaluation.**
>
> We conduct user study by inviting 40 participants and providing each of them with 30 randomly selected source images where the corresponding generated results of different methods are displayed randomly. Participants were asked to choose the image that better applies the requested edit while preserving most of the original image details. We newly added three SoTA methods DDS [Hertz A, et al., arXiv:2304.07090, 2023], InstructPix2Pix [Brooks T, et al., CVPR2023] and EDICT [Wallace B, et al., CVPR2023] for comparison. The percentage of votes for each method is shown in Tables r4-1 and r4-2. The results demonstrate that the participants exhibit a strong preference for our method.
>
> Table r4-1: User study results of Cat $\rightarrow$ Dog task.
>
> |ILVR|SDEdit|EGSDE|Ours|
> |:-:|:-:|:-:|:-:|
> |11.5%|10.5%|12.5%|**65.5%**|
>
> Table r4-2: User study results of the tasks on ImageNet, Imagen, COCO-S, and Dreambooth datasets.
>
> |SDEdit|DiffEdit|DDS|EDICT|InstructPix2Pix|Ours|
> |:-:|:-:|:-:|:-:|:-:|:-:|
> |4.5%|10.0%|3.0%|4.5%|6.0%|**72.0%**|
>
> **Q2: The method lacks novelty. Using different starting point with masked and non-masked region is just a variation of DiffEdit approach. Please elaborate the difference between baseline models.**
>
> First, our method and the DiffEdit method have different motivations. DiffEdit automatically generates a mask to indicate the regions of the input image that need to be edited, by contrasting predictions of a diffusion model conditioned on different text prompts. Then use the inferred mask to replace the background with pixel values from the encoding process at the corresponding timestep. Differently, we add noise to each pixel based on its relevance to the editing task to achieve controllable editing. We establish the relationship between the noise variance and the timestep through rigorous theoretical deducing, and propose to denoise at different timesteps for different pixels to utilize the pre-trained isotropic Gaussian diffusion model.
>
> Second, in terms of algorithm implementation, we are in a soft way compared with DiffEdit which uses the hard mask to guide the denoising process. Our algorithm starts the denoising process for a few pixels, and then gradually more pixels are included in the denoising process. Each pixel takes the denoising timestep according to its relevance to the editing task. This helps to generate more natural images and avoids artifacts caused by hard masks.
>
> Finally, in terms of generation quality, our method can generate images with more natural appearance. DiffEdit may generate unsmooth images around the boundary of the mask. For example, the images generated by DiffEdit shown in the first and fourth columns of Figure 3 in the main text have unnatural combination of the background and foreground.
>
> We will clarify more on these differences in revision.

---

> > ### Comment · Reviewer_rj2K · 2023-08-18
> >
> > The author rebuttal addressed my concerns, I keep my original score.

---

> > > ### Author Response · Authors · 2023-08-18
> > > **Thank Reviewer rj2K for the comments.**
> > >
> > > Thanks, and we will carefully revise the paper according to these questions and comments in the reviews.

---

### Official Review · Reviewer_Cf2x · 2023-07-05

**Soundness:** 3 good
**Presentation:** 2 fair
**Contribution:** 3 good
**Rating:** 5
**Confidence:** 4

**Summary:**

The authors proposed a Non-isotropic Gaussian Diffusion framework, which they used for Image-to-Image translation and editing, which apparently are basically “soft inpainting” tasks. The authors crafted a non-uniform version of regular gaussian diffusion but ultimately used some knowledge from it to drive a regular gaussian diffusion model. The method looks very similar to a naive inpaiting method, with some notable differences.

**Strengths:**

The paper shows a framework for image-editing, which is an increasingly important task for practical deployment. The authors also justified their method with some theretical backing, which is good.

Some good quantitative and qualitative results are shown.

**Weaknesses:**

Despite the method being overall sound, I feel it is explained with more complication than necessary. The crux of the method is relatively simple than it looks.

Effectively, what the method is doing is the following: It creates a “soft mask” (as shown in Fig.1 of supplementary) $\Lambda_k = [ \lambda_k ]_k$ for a given image that dictates which pixel $k$ requires more change in order to accomplish the translation/editing task. A time endpoint $T_k$ is created for each pixel based on $\lambda_k$ — which is higher for pixels that require relatively more change during the generative process. Starting the generative process at $t=T$, the generation is “faked” (use of Eq. 11) till every pixel hits it’s own $T_k$, after that, the original de-noiser is used to “fill the gaps”.

- In light of the above explanation, it seems all the theoretical description about NGDM is sort of unnecessary and over-complicated. The NGDM isn’t really used in it’s true sense. All it does is figures out $T_k = \xi_k(t = T)$ — the expression Eq. 8 isn’t really necessary apart from it’s value at $t=T$. Essentially, the NGDM part is solely used to establish “how long to fake the generation for each pixel”.
- Carrying on the previous point, the expression of $\xi_k(t=T)$ is also arbitrarily related to the soft mask values $\lambda_k$. There are many possible ways to incorporate $\Lambda(\mathcal{I})$ into the forward SDE, opening design choices for Eq.6. Depending on how $\Lambda(\mathcal{I})$ is incorporated into the SDE, the expression in Eq.8 will change. In that sense, it’s not clear as to why this specific design was chosen. One can completely do away with the NGDM framework and use an arbitrary function that relates $T_k$ to $\lambda_k$ — what’s wrong with that?
- Section B.2 has a rather good explanation with the hard-soft weighting matrix comparison. This should’ve been in the main paper and the explanation should’ve been geared towards “in-paiting with soft wighting”.
- The really important part of this framework seems to be computation of the attention map $\mathcal{A}(\mathcal{I})$, which isn’t really a contribution of this paper (section 3.4). How costly is that, in terms of computation and relative to the core editing part ?
- Quantitave results are pretty limited, unlike qualitative ones.
- What’s the difference between the editing and image-to-image translation task? How do you incorporate “cat” and “dog” class into the translation task? Isn’t it related to text-based editing itself?

Overall, I like the method, but I think the paper should’ve been written in a very different way. I would like the authors to take inspiration from the RePaint [1] paper.

[1] https://arxiv.org/pdf/2201.09865.pdf

**Questions:**

Please see the weakness section for consolidated comments and questions.

**Limitations:**

Not much limitation is mentioned. Some failure case and their possible explanation is written in suppl. They should be in the main paper.

---

> ### Author Rebuttal · Authors · 2023-08-10
>
> **Q1: Necessity of theoretical description and Eqs. (6)&(8)**
>
> We clarify the necessity of theoretical description and Eqs. (6,8) as follows.
>
> Firstly, our motivation is to achieve controllable image editing by adding noise with different variances to different pixels of the image. This motivates us to construct the non-isotropic Gaussian diffusion model (NGDM).
>
> Secondly, to avoid retraining, we propose to use the pre-trained isotropic Gaussian diffusion model (IGDM) for achieving the data sampling for NGDM. The key challenge is how to formulate the relation between NGDM and IGDM. Section 3.2 analyzes the relation between the variance of noise in NGDM and the time step in IGDM. The deduced result in Eq. (8) establishes the transformation from noise variance to time step.
>
> With this theoretical foundation, we can first calculate the weighting matrix $\mathbf{\Lambda}(\mathcal{I})$ based on the input image $\mathcal{I}$, then derive the denoising time steps of each pixel based on Eq. (8). If we do away with NGDM and discard the Eq. (8), we have no theoretical guarantee for designing function that relates $T_k$ to $\lambda_k$. Our theoretical analysis and Eq.(8) in Lemma 1 build the theoretical support for the selection of the function with theoretical interpretation.
>
> **Q2: Including an explanation of the hard-soft weighting matrix comparison in the main paper with an explanation geared towards "in-painting with soft weighting".**
>
> We will include the hard-soft weighting matrix comparison in the main paper. Mask-guided image editing is similar to the inpainting task. Both tasks have the same problem that the boundary of mask is prone to be with incorrect artifacts. Our Algorithm 1 gradually increases the denoising region with the increase of the denoising steps in the diffusion. Each pixel begins to be denoised with the denoising time step according to its relevance to the editing task. This helps to generate natural avoiding artifacts caused by a hard mask.
>
> **Q3: Cost of computing the attention map.**
>
> We conduct experiments on the single NVIDIA GeForce RTX3090. For an image with resolution of 512 × 512, the computation time of the attention map $\mathbf{\Lambda}(\mathcal{I})$ takes about 2.2 seconds, and the computation time of the core editing part takes about 3.8 seconds. A complete edit to an image takes about 6 seconds in total.
>
> **Q4: Quantitative results are pretty limited, unlike qualitative ones.**
>
> We add two natural datasets (COCO-S and DreamBooth Dataset [Ruiz N, et al., CVPR2023]) and SoTA methods DDS [Hertz A, et al., arXiv:2304.07090, 2023], InstructPix2Pix [Brooks T, et al., CVPR2023] and EDICT [Wallace B, et al., CVPR2023] for more comparison. The quantitative results are shown in Table r3-1. It can be seen that our method outperforms competing methods by achieving a better trade-off between CLIPScore and LPIPS value.
>
> Table r3-1: Quantitative comparison on COCO-S and Dreambooth dataset.
>
> |Method|SDEdit|DiffEdit|DDS|EDICT|InstructPix2Pix|Ours ($a$=10.0,$b$=5.0)|Ours ($a$=10.0,$b$=6.0)|
> |:-:|:-:|:-:|:-:|:-:|:-:|:-:|:-:|
> |CLIPScore$\uparrow$ (COCO-S dataset)|28.58|30.31|30.48|30.77|31.28|**31.75**|31.45|
> |LPIPS$\downarrow$ (COCO-S dataset)|55.76|29.60|31.53|24.00|36.55|28.80|**23.43**|
> |CLIPScore$\uparrow$ (Dreambooth dataset)|19.64|19.88|**20.17**|19.70|19.64|19.81|19.75|
> |LPIPS$\downarrow$ (Dreambooth dataset)|59.82|27.47|33.26|22.52|40.50|24.86|**19.95**|
>
> We conduct user study by providing 40 participants with 30 randomly selected source images and the corresponding generated results of different methods are displayed randomly. Participants were asked to choose the image that better applies the requested edit while preserving most of the original image details. The percentage of votes for each method is shown in Tables r2-1 and r2-2. The results demonstrate that the participants exhibit a strong preference to our method.
>
> Table r3-2: User study of Cat $\rightarrow$ Dog task.
>
> |ILVR|SDEdit|EGSDE|Ours|
> |:-:|:-:|:-:|:-:|
> |11.5%|10.5%|12.5%|**65.5%**|
>
> Table r2-2: User study results of the tasks on ImageNet, Imagen, COCO-S, and Dreambooth datasets.
>
> |SDEdit|DiffEdit|DDS|EDICT|InstructPix2Pix|Ours|
> |:-:|:-:|:-:|:-:|:-:|:-:|
> |4.5%|10.0%|3.0%|4.5%|6.0%|**72.0%**|
>
> We test the computational efficiency of SINE on NVIDIA Tesla V100, and the remaining methods on NVIDIA GeForce RTX3090. It can be seen that our method is comparable to other methods in both computational time and memory cost.
>
> Table r3-4: Computational time and memory cost.
>
> |Method|SDEdit|DiffEdit|SINE|DDS|InstructPix2Pix|EDICT|Ours|
> |:-:|:-:|:-:|:-:|:-:|:-:|:-:|:-:|
> |Time per iteration (s)$\downarrow$|**3**|9|3480|46|12|648|6|
> |Memory (GB)$\downarrow$|10.0|**6.7**|28.0|16.7|18.0|13.8|**6.7**|
>
> **Q5: Difference between the editing and image-to-image translation task and the way to incorporate "cat" and "dog" classes into the translation task.**
>
> Image editing is to modify source images under specific guidance while the image-to-image translation aims to learn the mapping between two visual domains.
>
> For translation, the "cat" images are used in the forward process, and "dog" images are used to train the specific pre-trained diffusion model for generating dog images without using the text information.
>
> **Q6: The paper should’ve been written in a very different way by taking inspiration from [Lugmayr A, et al., CVPR2022].**
>
> We will cite the paper and take inspiration from the paper in the final version preparation, if accepted.
>
> **Q7: Discussing limitation. Including failure case and their possible explanation in the main paper.**
>
> A limitation of our method is that incorrect weighting matrix may lead to the failure of the method. Moreover, our method relies on a pre-trained diffusion model. Artifacts are produced when the desired edit involves generation failure cases of the underlying model. We will include limitations and failure cases in the main paper.

---

> > ### Comment · Reviewer_Cf2x · 2023-08-12
> > **Response #1 to rebuttal**
> >
> > Thanks for the clarifications and extra results. Some of them were helpful.
> >
> > However, my primary objection remains -- I still do not see how the non-isotropic gaussian theory is necessary. Even though you said ..
> >
> > > "If we do away with NGDM .. we have no theoretical guarantee for designing function that relates $T_k$ to $\lambda_k$"
> >
> > .. I do not see any theoretical guarantee here either. Even if there is, it's not quite clear in the paper or in your response.
> >
> > Also, I tend to agree with two other reviewers who said the qualitative results aren't clearly better as other methods do quite good. Yet, surprisingly, your user study subjects prefer your method with significantly high percentage !
> >
> > At the end, I will keep my BA rating, but at the same time I can see there are grounds for rejection.

---

> > > ### Author Response · Authors · 2023-08-13
> > > **Response to Reviewer Cf2x (Part 1/3)**
> > >
> > > Thanks for the comments. We would like to further clarify the necessity of non-isotropic Gaussian diffusion model (i.e., Eq. (6)) and the deduction of the relation between $T_k$ and $\lambda_k$ (i.e., Eq. (8)), and the detailed comparison of different methods as follows.
> > >
> > >
> > > **(1) Necessity of NGDM in Eq. (6).**
> > >
> > > In controllable image editing, our goal is to translate/edit a specific object/thing of an image, while preserving the remaining parts of the image. As empirically shown by the previous works [10,16], in the diffusion models, if adding larger scale (i.e., noise variance) noise to the image and then denoising it, it tends to largely modify the input image. While adding a smaller scale of noise to the input image and then denoising it will better preserve the information of the input image after editing. This motivates us to add noises with different variances to different pixels in diffusion models for controllable editing, to ensure that the image regions with larger noise variance will be changed more heavily in editing. This idea is formulated as the Non-isotropic Gaussian diffusion Model (NGDM) in Eq. (6).  In NGDM, we introduce a $\lambda_k\in [0,1]$ for each pixel in the forward VP-SDE to control the scale of the added noise and the degeneration of the image.
> > >
> > > **(2) Necessity of the relation between $T_k$ and $\lambda_k$ in Eq. (8).**
> > >
> > > Based on the NGDM in Eq. (6), we may train a score-based model for NGDM and then conduct the corresponding reverse SDE to obtain the edited image. However, to avoid the training of the Non-isotropic Gaussian diffusion model (NGDM), we utilize the existing pre-trained Isotropic Gaussian diffusion Model (IGDM) to realize the image sampling for the NGDM, which is the major contribution of this paper. Theorem 1 established the conclusion that the NGDM in Eq. (7) can be rectified to an IGDM model in Eq. (9) but with different total diffusion time $T_k$ for different pixel indexed by $k$, determined based on Eq. (8). This inspires us to utilize the pre-trained IGDM to achieve the data sampling of NGDM for image editing. Since the IGDM in Eq. (9) has different total diffusion time $T_k$ for different pixel $k$, in the reverse process of IGDM, we should set different starting time, i.e., $T_k$, to different pixel $k$ for denoising, using the pre-trained IGDM. And the corresponding data sampling algorithm is in Algorithm 1. In summary, the NGDM, Theorem 1, and Eq. (8) motivate and guide us to design the image sampling algorithm 1.
> > >
> > > If we do away with the NGDM & Eq. (8) and focus on heuristically designing the relation between $T_k$ and $\lambda_k$, we would face the following problems. (I) It is unclear how to design the relation between $T_k$ and $\lambda_k$. Actually, there even lacks a good motivation to determine whether $T_k$ and $\lambda_k$ are positively or inversely correlated. (II) The formulation of the relation between $T_k$ and $\lambda_k$ is unclear. Even though we can empirically choose it, the explanation is lacking. (III) The motivation and guidance for the design of the image sampling algorithm, i.e., different pixels should start at different times in the reverse diffusion process, are unclear.
> > >
> > > **(3) Comparison with examples of heuristic designed relation between $T_k$ and $\lambda_k$.**
> > >
> > > To further testify whether the heuristic designed relation between $T_k$ and $\lambda_k$ works, we compare our approach with the following examples of the heuristic designs: $T_k=\lambda_kT$, $T_k=\lambda_k^2T$, $T_k=\sqrt{\lambda_k}T$.
> > >
> > > Table r-2-1: Results for different designs of the relation between $T_k$ and $\lambda_k$ on image editing task on COCO-S dataset.
> > >
> > > ||$T_k=\lambda_kT$|$T_k=\lambda_k^2T$|$T_k=\sqrt{\lambda_k}T$|Ours|
> > > |:-:|:-:|:-:|:-:|:-:|
> > > |CLIPScore$\uparrow$|31.22|28.78|31.47|**31.75**|
> > > |LPIPS$\downarrow$|31.30|**27.55**|29.12|28.80|
> > >
> > > From Table r-2-1, we can see that our approach achieves the best CLIPScore and second best LPIPS. The $T_k=\lambda_k^2T$ achieves the best LPIPS but an obviously lower CLIPScore than the other approach. Correspondingly, we find that $T_k=\lambda_k^2T$ does not successfully modify the object in the images to the desired one to accomplish the editing task. The result of $T_k=\sqrt{\lambda_k}T$ better approaches that of our method, which may be because $\sqrt{\lambda_k}T$ is close to the $\xi_k(T)$ in Eq. (8) (in Eq. (8), if we set $\beta_{\rm min}=0$, we have $\xi_k(T)=\sqrt{\lambda_k}T$). Nevertheless, as discussed in **(2)**, we do not have the motivation for the choice of $\sqrt{\lambda_k}T$.

---

> > > ### Author Response · Authors · 2023-08-13
> > > **Response to Reviewer Cf2x (Part 2/3)**
> > >
> > > **(4) Comparison of different methods qualitatively.**
> > >
> > > We focus on controllable image translation and image editing that aim to modify the image regions related to the task, while leaving the other regions unchanged to preserve the structure/details of the source image as much as possible. We next clarify that our approach better achieves this goal qualitatively.
> > >
> > > In Figure 2 in the main paper for cat->dog translation, it can be observed that the regions (background) of the source images outside the cat are preserved in the translated images (the 2nd row) by our approach. While the other approaches can not always preserve the regions of the source images outside the cat. For example, (1) *in the 2nd column*, the green grass on the background wall of the source images disappeared in the translated images by the EGSDE & SDEit & ILVR, but our approach keeps the green grass; (2) *In the 5th column*, the white stones in the source image was preserved by our approach, but EGSDE & SDEit & ILVR produce white blurred things which are not stones. (3) *In the 6th column*, the pose of the cat in the source image is preserved when the cat is translated to be the dog by our method, but the poses of the cats in the images generated by the other methods are changed. Note that for the other columns, our method can also better preserve the regions outside the cat. Please zoom in on the figure to see these differences between the translated images.
> > >
> > > In Figure 3 in the main paper for translation on ImageNet dataset, it can be observed that our method (the 2nd row) can always accurately translate the category of the source image into the category given by the target prompt, while maintaining the original image regions that are not related to translation task. While the other methods may fail to translate or fail to maintain information irrelevant to editing in some cases. For example, (1) *in the first column*, DiffEdit produces artifacts when editing oystercatcher into flamingo, which can be seen from the black area in the middle part of the generated image. DiffuseIT does not successfully edit the oystercatcher into flamingo. DDIB and SDEdit do not maintain the background region outside the oystercatcher in the source image. (2) *In the 3rd column*, DiffEdit & DiffuseIT & DDIB & SDEdit do not preserve the regions of the source images outside the convertible, but our method preserves them well. (3) *In the 4th column*, the boundary of the lemon in the generated image by DiffEdit has artifacts. DiffuseIT & DDIB & SDEdit do not preserve the regions of the source images outside the custard apple. However, our method can preserve most of the details that are irrelevant to editing. (4) *In the 6th column*, DiffEdit & DDIB & SDEdit do not preserve the tree below the kite in the source image. DiffuseIT does not generate bald eagle based on the target prompt. Note that for the other columns, our method also accurately translates the category of the source image into the category given by the target prompt, while preserving information that is not related to translation. Please zoom in on the figure to see these differences between the translated images.
> > >
> > > In Figure 4 in the main paper for image editing on Imagen dataset, our method can successfully edit the source image based on the target prompt while making minimal modifications to the source image. For example, (1) *in the first column*, SINE does not edit the beach into the mountain. The appearance of the panda in the generated image by SDEdit changes compared to the panda in the source image. (2) *In the 2nd and 3rd columns*, DiffEdit and SDEdit do not preserve the background outside the animal in the source image. (3) *In the 4th column*, SINE does not edit the mountain into the beach and SDEdit changes the appearance of cat while editing. (4) *In the 5th column*, images generated by DiffEdit and SDEdit can not preserve the appearance of the cat, while SINE can not edit sunglasses into hat. Note that for the other columns, our method can also successfully edit based on prompt, while preserving information that is not related to editing. Please zoom in on the figure to see these differences between the translated images.

---

> > > ### Author Response · Authors · 2023-08-13
> > > **Response to Reviewer Cf2x (Part 3/3)**
> > >
> > > In Figure r1 in the uploaded ong-page pdf (in the attachment of official comment with title "Author Rebuttal by Authors") for image editing on COCO-S and Dreambooth dataset, our method can always edit the source image based on the target prompt, while maintaining the information irrelevant to editing unchanged, compared with the SoTA method. For example, (1) *in the first column*, DiffEdit & SDEdit & InstructPix2Pix & EDICT do not generate images that match the target text. Moreover, DiffEdit & SDEdit & DDS & InstructPix2Pix & EDICT can not preserve the detailed background outside the bird. (2) *In the 2nd column*, DDS & InstructPix2Pix & EDICT do not generate luggage based on the target prompt, and DiffEdit & SDEdit & DDS & InstructPix2Pix & EDICT can not preserve the regions that are not related to editing. (3) *In the 5th column*, DDS & InstructPix2Pix & EDICT do not change the background into mountain and SDEdit does not preserve the stuffed animal in the source image. (4) *In the 6th column*, the cats in the generated images by SDEdit & DDS & EDICT do not wear a rainbow scarf. The regions below the cat in the images generated by DiffEdit & SDEdit & DDS & InstructPix2Pix & EDICT are not similar to the corresponding region in the source image. Our method not only generates image of a cat wearing a rainbow scarf, but also preserves the detailed background below the cat. Note that for the other columns, our method can also successfully edit based on prompt, while preserving information that is not related to editing. Please zoom in on the figure to see these differences between the translated images.
> > >
> > > In summary, our method can not only successfully edit the source image based on the target prompt, but also keep the information irrelevant to editing in the source image unchanged.
> > >
> > >
> > >
> > > **(5) About the user study in the responses.**
> > >
> > > For the user study, we queried 40 participants to score 30 groups of randomly selected source images and the corresponding generated results by different methods. The generated images of ours and the other methods are displayed randomly in order, and the participants do not know the methods corresponding to these generated images. Participants are suggested to select the best result that better applies the requested edit while minimally modifying the source image. For Cat $\rightarrow$ Dog translation task, we set the question: "Which image below better translates cat into dog, while minimally modifying the source image?" For the text-guided image editing task, we set the question: "Which image below better applies the requested edit to the source image on top, while minimally modifying the source image?" From the results of the user study, most participants favored our method. As analyzed based on the qualitative results above, our approach can better preserve the image regions outside the regions related to the editing task that should be modified. In the user study, the participants would like to choose the successfully translated images that are minimally changed from the source image, which is consistent with the qualitative results.

---

### Official Review · Reviewer_Jr5M · 2023-07-18

**Soundness:** 3 good
**Presentation:** 3 good
**Contribution:** 3 good
**Rating:** 5
**Confidence:** 3

**Summary:**

The paper proposes a non-isotropic gaussian diffusion model, in contrast to current popular isotropic gaussian diffusion models. The paper also proposes new forward and reverse diffusion processes in accordance to the non-isotropic gaussian corruption framework that is proposed. The motivation behind using non-isotropic gaussian noise is that "the diffusion model can generate more diverse novel content if adding noise with larger variance to the image while preserving the image content if adding smaller variance noise". Experimental results on image editing show superior quantitative performance for the proposed method.

**Strengths:**

- The idea of using non-isotropic gaussians for diffusion corruptions seems novel, to the best of my knowledge.
- The forward and backward processes make sense and seem to work well enough to achieve similar performance to vanilla isotropic models.
- The design allows for use of isotropic models which makes this a flexible method that can be easily inserted in many applications without retraining the model, which is very positive.
- The main strength seems to be that the method is able to insert a different level of detail in different parts of the image, which improves editing (since different parts of the image have to be edited in different ways to achieve strong results). For example some parts of the image should be largely preserved while others should be heavily structurally changed. Once can see this in Fig. 3 with the custard apple -> lemon example.

**Weaknesses:**

- It's hard to see, on average, a large perceptual difference and superiority of the method in the qualitative translation figures. Maybe pointing to critical regions would help, but on average it seems like other methods are not too bad. Also, how were the samples for figures selected?
- A user study on a large amount of data would clarify how users perceive these changes.
- The method does achieve low LPIPS for a large CLIP score, but DiffEdit is close in relative terms.
- The paper focuses on image editing but the title does not include the term. Would be good to include to be more specific.


Some related work that could potentially be included (not mine):

Bansal, Arpit, et al. "Cold diffusion: Inverting arbitrary image transforms without noise." arXiv preprint arXiv:2208.09392 (2022).

Daras, Giannis, et al. "Soft diffusion: Score matching for general corruptions." arXiv preprint arXiv:2209.05442 (2022).

**Questions:**

I think the paper is well presented and I understood it. The experiments are well laid out. The question would be to directly try to rebut the weaknesses I claim, and I will take into account other reviews to see whether I am being too harsh with respect to the relevance of the effect size on experiments (and the breadth of tasks tackled).

**Limitations:**

I think the main limitation is the motivation and experiments. Although the method is interesting, and the theoretical motivation (different parts of the image need to be treated slightly differently) is compelling, the final results don't seem to support the motivation so strongly and are isolated to the editing task. Maybe tackling more tasks with such a general methodology could be compelling? Or a user study that shows that users prefer these images, with large effect size?

---

> ### Author Rebuttal · Authors · 2023-08-10
>
> **Q1: On the performance improvement over baselines and the way to select visualization samples**
>
> We tackle controllable image editing that modifies the image regions related to the editing task while leaving the other regions unchanged. As shown in Figure 2 in the paper, our method can better preserve the background, pose, etc., compared with other methods. For example, Figure 2 shows that for images in columns 2-5 with relatively complex backgrounds, our method can accurately keep the backgrounds of the source images unchanged while translating cats into dogs. Other methods either blur the backgrounds or fail to maintain the source image backgrounds correctly. In Figure 3, the source images in columns 2 and 4 have complex backgrounds, and our method can better preserve these details when editing images. Besides the metrics reported in the paper, we also conduct user study in the rebuttal, which shows an obvious performance improvement achieved by our method (please refer to Q2).
>
> These displayed examples are randomly selected from each dataset, typically with source images having various poses and backgrounds or with various editing types (such as background replacement or object transformation). Due to space limit, we can only show a few examples, but overall these conclusions are the same in the other examples of these images in the datasets. We also randomly selected several failure examples in Figure 6 of Appendix.
>
> **Q2: A user study on a large amount of data.**
>
> We conduct user study by providing 40 participants with 30 randomly selected source images and the corresponding generated results of different methods are displayed randomly. Participants were asked to choose the image that better applies the requested edit while preserving most of the original image details. The percentage of votes for each method is shown in Tables r2-1 and r2-2. The results demonstrate that the participants exhibit a strong preference for our method.
>
> Table r2-1: User study results of Cat $\rightarrow$ Dog task.
>
> |ILVR|SDEdit|EGSDE|Ours|
> |:-:|:-:|:-:|:-:|
> |11.5%|10.5%|12.5%|**65.5%**|
>
> Table r2-2: User study results of the tasks on ImageNet, Imagen, COCO-S, and Dreambooth datasets.
>
> |SDEdit|DiffEdit|DDS|EDICT|InstructPix2Pix|Ours|
> |:-:|:-:|:-:|:-:|:-:|:-:|
> |4.5%|10.0%|3.0%|4.5%|6.0%|**72.0%**|
>
> **Q3: The method does achieve low LPIPS for a large CLIP score, but DiffEdit is close in relative terms.**
>
> DiffEdit is close to our method on CLIPScore and LPIPS metrics. However, from a qualitative comparison point of view as shown in Figure 3, our method can generate more natural images, while DiffEdit is prone to produce boundary artifacts caused by hard mask, resulting in an unsmooth combination of foreground and background. Please refer to the images in the first and fourth columns of Figure 3 for the examples. This may not be reflected by the CLIPScore and LPIPS. Note that the results of user study in Table r2-2 show that the participants prefer the results of our method.
>
> **Q4: Including "image editing" in title.**
>
> Thanks for this suggestion, we will consider changing the title to "Constructing Non-isotropic Gaussian Diffusion Model Using Isotropic Gaussian Diffusion Model for Image Editing", as suggested.
>
> **Q5: Including the related works [Bansal A, et al., arXiv:2208.09392, 2022] and [Daras G, et al., arXiv:2209.05442, 2022.]**
>
> We will include these references in the Introduction section. Cold Diffusion [Bansal A, et al., arXiv:2208.09392, 2022] is based on generalized diffusion models that were built on arbitrary image transformations like blurring, downsampling, etc. And a trained restoration network is created to perform denoising. Soft diffusion [Daras G, et al., arXiv:2209.05442, 2022.] is based on general linear corruption processes and it learns the diffusion model by training objective of Soft Score Matching for the linear corruption process. Differently, we consider non-isotropic Gaussian noise and utilize a pre-trained isotropic Gaussian diffusion model to achieve sampling without retraining.
>
> **Q6: More experimental tasks and user study to support motivation.**
>
> Image editing task aims at modifying source images under the target prompt guidance while leaving the regions that are unrelated to editing unchanged, to generate images that are as similar as possible to source image. It is empirically known that the diffusion model can generate more diverse novel content if adding noise with a larger variance to the image while preserving the image information if adding smaller variance noise. Motivated by this, we employ a non-isotropic diffusion model to add noises with different variances to different image pixels. We achieve controllable editing by adding different variance noises to different image pixels, with varying noise variance considering the degree to which the corresponding pixels should be edited/preserved. We will explain this motivation for using NGDM for controllable image editing in more detail in the Introduction section.
>
> **More tasks.** We add experiments on two additional tasks that are not covered in the experimental section, including local style transfer and gender transformation. The local style transfer task aims to transform the specified object in the image into another style without changing the structure, while preserving the information of the rest of the region. For example, transforming a "real dog" into a "sculptural dog", while keeping the background unchanged. Gender transformation aims to turn males into females while keeping the structure of the face unchanged. Due to space limit, we show a few examples in Figure r2 of the uploaded one-page pdf, and we will show more examples in Appendix B in the revised version.
>
> **User study.** As suggested, we report the results of user study in Tables r2-1 and r2-2. The results show that the participants exhibit a strong preference for our method.

---

> > ### Author Response · Authors · 2023-08-17
> > **Further clarification on qualitative comparisons and user study**
> >
> > Dear reviewer, thanks for your comments and questions. Besides the rebuttal, in the author-reviewer discussion phase, we additionally clarified the qualitative comparisons and user studies in the official comment with title "To ACs and Reviewers: further clarification on the qualitative comparison with other methods and on the user study in the responses.", following the "Author Rebuttal by Authors".  Please refer to them for detailed clarifications.

---

> > ### Comment · Reviewer_Jr5M · 2023-08-17
> >
> > Thank you. The addition of (large enough) user studies and the careful rebuttal, along with the comments from other reviewers have convinced me to increase my score to 5.

---

> > > ### Author Response · Authors · 2023-08-18
> > > **Thank Reviewer Jr5M for the positive comments.**
> > >
> > > Thanks, and we will carefully revise the paper according to these questions and comments in the reviews.

---

### Official Review · Reviewer_xHHG · 2023-07-29

**Soundness:** 3 good
**Presentation:** 3 good
**Contribution:** 3 good
**Rating:** 6
**Confidence:** 3

**Summary:**

The authors proposed a Non-isotropic Gaussian Diffusion Model for the task of image-image translation and image editing. The NGDM is achieved by adding different noise variances to different image pixels so as to control the regions to edit. Experimental results have demonstrated the state-of-the-art quality of the proposed paper.

**Strengths:**

- The paper proposed a practical method for image-image translation/editing by utilizing the off-the-shelf diffusion model. It seems like the proposed method is easy to re-implement and has demonstrated satisfactory results.

- The presentation of the paper is good.

**Weaknesses:**

- Why the proposed method would choose to implement the non-isotropic diffusion process by controlling each pixel's denoising steps? I can come up with one alternative quickly: at each iteration, we can add different levels of noise to different pixels (by also using the input-dependent weight matrix), and denoise as usual. It is a bit strange to set different denoise time steps for different pixels.

- I would like to see comparisons with the recent paper "Delta Denoising Score", which also targets the task of controllable image editing.


**Questions:**

See above

**Limitations:**

See above

---

> ### Author Rebuttal · Authors · 2023-08-10
>
> **Q1: The reason for implementing NGDM by controlling each pixel’s denoising steps.**
>
> It is empirically known that the diffusion model can generate more diverse novel content if adding noise with a larger variance to the image while preserving the image information if adding smaller variance noise. Motivated by this, we employ a non-isotropic Gaussian diffusion model to add noises with different variances to different image pixels for controllable image editing.
>
> As suggested by the reviewer, we can directly add different levels of noise to different pixels, and denoise as usual using pre-trained diffusion models. Actually, we have tried this strategy when working on this paper, but only produced noises rather than images. The main reason could be that the transition kernel at time $t$ of the isotropic Gaussian diffusion model is different from that of the non-isotropic Gaussian diffusion model, which is related to the weighting matrix that determines the variance. Therefore, the score under isotropic Gaussian diffusion model at time step $t$ does not match the score under the non-isotropic Gaussian diffusion model. The pre-trained isotropic Gaussian diffusion model will produce incorrect score predictions at time $t$ for non-isotropic Gaussian diffusion model.
>
> One way is to retrain the non-isotropic Gaussian diffusion model. We instead use the off-the-shelf pre-trained isotropic Gaussian diffusion model for achieving the date sampling to avoid retraining. To implement this idea, we use Lemma 1 in the main paper to establish a relationship between the noise level in the non-isotropic Gaussian diffusion model and the time step $t$ in the isotropic Gaussian diffusion model. We prove in Theorem 1 that the transition kernel at time $t$ in the non-isotropic Gaussian diffusion model is equal to the transition kernel at time step $\tau$ in the isotropic Gaussian diffusion model, where $\tau$ depends on the noise level in the non-isotropic Gaussian diffusion model. Finally, since isotropic Gaussian diffusion model needs to accept the entire image as input, we propose a sampling algorithm of NGDM (Algorithm 1) to generate images using pre-trained isotropic Gaussian diffusion model.
>
>
> **Q2: Comparisons with "Delta Denoising Score".**
>
> DDS [Hertz A, et al., arXiv:2304.07090, 2023.] utilizes score distillation sampling mechanism for image editing. The authors optimize to produce the edited image given a text description by distilling the output of the score-based model on the reference source image-text pair. Differently, we aim to edit the image and leave the region unrelated to target prompt unchanged. We achieve this goal using a non-isotropic Gaussian diffusion model that adds independent noise with different variances to different pixels. We further rectify it into an isotropic Gaussian diffusion model with different pixels having different total forward diffusion times, for utilizing the pre-trained isotropic models for sampling.
>
> We conduct experiments on the ImageNet dataset and Imagen dataset introduced in Section 4 of the paper. In the rebuttal, we additionally consider two datasets (COCO-S and DreamBooth Dataset [Ruiz N, et al., CVPR2023]) for image editing task.
>
> We show qualitative visual results on COCO-S and Dreambooth datasets in Figure r1 of the uploaded one-page pdf. We report the CLIPScore and LPIPS metric in Table r1-1, the results of user study in Table r1-2, and the computational time and memory cost in Table r1-3. Table r1-1 shows that our method consistently achieves the best results on the four datasets. A larger CLIPScore denotes better alignment with the target text, while a smaller LPIPS value suggests higher fidelity to the source image. Compared with DDS, our method achieves a smaller LPIPS distance with a larger CLIPScore, which shows that our method can make smaller changes to the source image when editing.
>
> Table r1-1: CLIPScore ($\uparrow$) and LPIPS ($\downarrow$) on four datasets.
> |Method|Imagenet (CLIPScore)|Imagenet (LPIPS)|Imagen (CLIPScore)|Imagen (LPIPS)|COCO-S (CLIPScore)|COCO-S (LPIPS)|Dreambooth (CLIPScore)|Dreambooth (LPIPS)|
> |:-:|:-:|:-:|:-:|:-:|:-:|:-:|:-:|:-:|
> |DDS|28.13|43.57|34.69|31.38|30.48|24.00|**20.17**|22.52|
> |Ours|**30.66**|**31.32**|**35.66**|**20.28**|**31.75**|**23.43**|19.81|**19.95**|
>
> We conduct user study by providing 40 participants with 30 randomly selected source images and the corresponding generated results of different methods are displayed randomly. Participants were asked to choose the image that best achieves the requested edit while preserving most of the original image details. The percentage of votes for each method is shown in Table r1-2. The results demonstrate that the participants exhibit a strong preference for our method.
>
> Table r1-2: User study results.
>
> |SDEdit|DiffEdit|DDS|EDICT|InstructPix2Pix|Ours|
> |:-:|:-:|:-:|:-:|:-:|:-:|
> |4.5%|10.0%|3.0%|4.5%|6.0%|**72.0%**|
>
> We test the computational efficiency on NVIDIA GeForce RTX3090. From Table r1-3, it can be seen that the running time of DDS is about 7.7 times that of ours and the memory occupied by DDS is also higher than ours.
>
> Table r1-3: Computational time and memory cost.
>
> |Method|DDS|Ours|
> |-|:-:|:-:|
> |Time per iteration (s)$\downarrow$|46|**6**|
> |Memory (GB)$\downarrow$|16.7|**6.7**|

---

> ### Comment · Area_Chair_6fEM · 2023-08-18
>
> Reviewer xHHG: Your review is very thin and you have not responded to the authors rebuttal. I will not be able to take your review into account unless you engage more meaningfully with this paper.

---

### Author Rebuttal · Authors · 2023-08-10

Dear ACs and reviewers,

Thanks for the insightful comments and suggestions on our paper. We have carefully responded to the comments of each reviewer. Meanwhile, we have uploaded a PDF file to show visual results as support material. We will revise our paper accordingly in the final version if accepted.

Best,

Authors

---

> ### Author Response · Authors · 2023-08-13
> **To ACs and Reviewers (Part 1/2): further clarification on the qualitative comparison with other methods and on the user study in the responses.**
>
> For a better understanding of the performance of our approach, we further clarify the qualitative comparison with other methods and the user study in the responses as follows.
>
> **(1) Further clarification on the qualitative comparison of different methods.**
>
> We focus on controllable image translation and image editing that aim to modify the image regions related to the task, while leaving the other regions unchanged to preserve the structure/details of the source image as much as possible. We next clarify that our approach better achieves this goal qualitatively.
>
> In Figure 2 in the main paper for cat->dog translation, it can be observed that the regions (background) of the source images outside the cat are preserved in the translated images (the 2nd row) by our approach. While the other approaches can not always preserve the regions of the source images outside the cat. For example, (1) *in the 2nd column*, the green grass on the background wall of the source images disappeared in the translated images by the EGSDE & SDEit & ILVR, but our approach keeps the green grass; (2) *In the 5th column*, the white stones in the source image was preserved by our approach, but EGSDE & SDEit & ILVR produce white blurred things which are not stones. (3) *In the 6th column*, the pose of the cat in the source image is preserved when the cat is translated to be the dog by our method, but the poses of the cats in the images generated by the other methods are changed. Note that for the other columns, our method can also better preserve the regions outside the cat. Please zoom in on the figure to see these differences between the translated images.
>
> In Figure 3 in the main paper for translation on ImageNet dataset, it can be observed that our method (the 2nd row) can always accurately translate the category of the source image into the category given by the target prompt, while maintaining the original image regions that are not related to translation task. While the other methods may fail to translate or fail to maintain information irrelevant to editing in some cases. For example, (1) *in the first column*, DiffEdit produces artifacts when editing oystercatcher into flamingo, which can be seen from the black area in the middle part of the generated image. DiffuseIT does not successfully edit the oystercatcher into flamingo. DDIB and SDEdit do not maintain the background region outside the oystercatcher in the source image. (2) *In the 3rd column*, DiffEdit & DiffuseIT & DDIB & SDEdit do not preserve the regions of the source images outside the convertible, but our method preserves them well. (3) *In the 4th column*, the boundary of the lemon in the generated image by DiffEdit has artifacts. DiffuseIT & DDIB & SDEdit do not preserve the regions of the source images outside the custard apple. However, our method can preserve most of the details that are irrelevant to editing. (4) *In the 6th column*, DiffEdit & DDIB & SDEdit do not preserve the tree below the kite in the source image. DiffuseIT does not generate bald eagle based on the target prompt. Note that for the other columns, our method also accurately translates the category of the source image into the category given by the target prompt, while preserving information that is not related to translation. Please zoom in on the figure to see these differences between the translated images.
>
> In Figure 4 in the main paper for image editing on Imagen dataset, our method can successfully edit the source image based on the target prompt while making minimal modifications to the source image. For example, (1) *in the first column*, SINE does not edit the beach into the mountain. The appearance of the panda in the generated image by SDEdit changes compared to the panda in the source image. (2) *In the 2nd and 3rd columns*, DiffEdit and SDEdit do not preserve the background outside the animal in the source image. (3) *In the 4th column*, SINE does not edit the mountain into the beach and SDEdit changes the appearance of cat while editing. (4) *In the 5th column*, images generated by DiffEdit and SDEdit can not preserve the appearance of the cat, while SINE can not edit sunglasses into hat. Note that for the other columns, our method can also successfully edit based on prompt, while preserving information that is not related to editing. Please zoom in on the figure to see these differences between the translated images.

---

> ### Author Response · Authors · 2023-08-13
> **To ACs and Reviewers (Part 2/2): further clarification on the qualitative comparison with other methods and on the user study in the responses.**
>
> In Figure r1 in the uploaded ong-page pdf (in the attachment of official comment with title "Author Rebuttal by Authors") for image editing on COCO-S and Dreambooth dataset, our method can always edit the source image based on the target prompt, while maintaining the information irrelevant to editing unchanged, compared with the SoTA method. For example, (1) *in the first column*, DiffEdit & SDEdit & InstructPix2Pix & EDICT do not generate images that match the target text. Moreover, DiffEdit & SDEdit & DDS & InstructPix2Pix & EDICT can not preserve the detailed background outside the bird. (2) *In the 2nd column*, DDS & InstructPix2Pix & EDICT do not generate luggage based on the target prompt, and DiffEdit & SDEdit & DDS & InstructPix2Pix & EDICT can not preserve the regions that are not related to editing. (3) *In the 5th column*, DDS & InstructPix2Pix & EDICT do not change the background into mountain and SDEdit does not preserve the stuffed animal in the source image. (4) *In the 6th column*, the cats in the generated images by SDEdit & DDS & EDICT do not wear a rainbow scarf. The regions below the cat in the images generated by DiffEdit & SDEdit & DDS & InstructPix2Pix & EDICT are not similar to the corresponding region in the source image. Our method not only generates image of a cat wearing a rainbow scarf, but also preserves the detailed background below the cat. Note that for the other columns, our method can also successfully edit based on prompt, while preserving information that is not related to editing. Please zoom in on the figure to see these differences between the translated images.
>
> In summary, our method can not only successfully edit the source image based on the target prompt, but also keep the information irrelevant to editing in the source image unchanged.
>
>
> **(2) Further clarification on the user study in the responses.**
>
> For the user study, we queried 40 participants to score 30 groups of randomly selected source images and the corresponding generated results by different methods. The generated images of ours and the other methods are displayed randomly in order, and the participants do not know the methods corresponding to these generated images. Participants are suggested to select the best result that better applies the requested edit while minimally modifying the source image. For Cat $\rightarrow$ Dog translation task, we set the question: "Which image below better translates cat into dog, while minimally modifying the source image?" For the text-guided image editing task, we set the question: "Which image below better applies the requested edit to the source image on top, while minimally modifying the source image?" From the results of the user study, most participants favored our method. As analyzed based on the qualitative results above, our approach can better preserve the image regions outside the regions related to the editing task that should be modified. In the user study, the participants would like to choose the successfully translated images that are minimally changed from the source image, which is consistent with the qualitative results.

---

### Decision · Program_Chairs · 2023-09-21

**Decision:**

Accept (poster)

**Comment:**

The paper presents a method for image-to-image translation and image editing called Non-isotropic Gaussian Diffusion Model (NGDM). NGDM is a score-based diffusion model that can be used to generate images that are both faithful to the source image and aligned with the desired translation/editing target. NGDM is constructed by adding independent Gaussian noises with different variances to different image pixels. The paper shows that NGDM can be used to achieve impressive results on both image-to-image translation and image editing tasks.

While the reviewers are not in uniform agreement that the paper should be published, I believe the authors have done an adequate and thorough job in addressing all the concerns, to the extent where no major concerns about the overall merit of the work should remain. Overall, I judge the paper to be a sufficiently strong contribution to be published.